# Shortest Path Networks for Graph Property Prediction

**Ralph Abboud, Radoslav Dimitrov, İsmail İlkan Ceylan**
Department of Computer Science
University of Oxford, UK
`firstname.lastname@cs.ox.ac.uk`

## Abstract

Most graph neural network models rely on a particular message passing paradigm, where the idea is to iteratively propagate node representations of a graph to each node in the *direct neighborhood*. While very prominent, this paradigm leads to *information propagation bottlenecks*, as information is repeatedly compressed at intermediary node representations, which causes loss of information, making it practically impossible to gather meaningful signals from distant nodes. To address this, we propose *shortest path message passing neural networks*, where the node representations of a graph are propagated to each node in the *shortest path neighborhoods*. In this setting, nodes can directly communicate between each other even if they are not neighbors, breaking the information bottleneck and hence leading to more adequately learned representations. Our framework generalizes message passing neural networks, resulting in a class of more expressive models, including some recent state-of-the-art models. We verify the capacity of a basic model of this framework on dedicated synthetic experiments, and on real-world graph classification and regression benchmarks, and obtain state-of-the-art results.

## 1 Introduction

Graphs provide a powerful abstraction for relational data in a wide range of domains, ranging from systems in life-sciences (e.g., physical [1, 2], chemical [3, 4], and biological systems [5, 6]) to social networks [7], which sparked interest in machine learning over graphs. Graph neural networks (GNNs) [8, 9] have become prominent models for graph machine learning, owing to their adaptability to different graphs, and their capacity to explicitly encode desirable relational inductive biases [10], such as permutation invariance (resp., equivariance) relative to graph nodes.

The vast majority of GNNs [11–13] are instances of *message passing neural networks (MPNNs)* [14], since they follow a specific message passing paradigm, where each node iteratively updates its state by aggregating messages from its *direct neighborhood*. This mode of operation, however, is known to lead to *information propagation bottlenecks* when the learning task requires interactions between distant nodes of a graph [15]. In order to exchange information between nodes which are $k$ hops away from each other in a graph, at least $k$ message passing iterations (or, equivalently, $k$ network layers) are needed. For most graphs, however, the number of nodes in each node's receptive field can grow exponentially in $k$. Eventually, the information from this exponentially-growing receptive field is compressed into fixed-length node state vectors, which leads to a phenomenon referred to as *over-squashing* [15], causing a severe loss of information as $k$ increases.

Several message passing techniques have been proposed to allow more global communication between nodes. Multi-hop models [16, 17], based on powers of the graph adjacency matrix, and transformer-based models [18–20] employing full pairwise node attention, look beyond direct neighborhoods, but both suffer from noise and scalability limitations. More recently, several approaches have refined message passing using *shortest paths* between pairs of nodes, such that nodes interact differently based on the minimum distance between them [21–23]. Models in this category, such as Graphormer [23], have in fact achieved state-of-the-art results. However, the theoretical study of this message passing paradigm remains incomplete, with its expressiveness and propagation properties left *unknown*.

R. Abboud, R. Dimitrov, İ. İ. Ceylan, Shortest Path Networks for Graph Property Prediction. *Proceedings of the First Learning on Graphs Conference (LoG 2022)*, PMLR 198, Virtual Event, December 9–12, 2022.

In this paper, we introduce *shortest path message passing neural networks (SP-MPNNs)* and study the properties of the models in this framework. The core idea behind this framework is to update node states by aggregating messages from *shortest path neighborhoods* instead of the *direct neighborhood*. Specifically, for each node $u$ in a graph $G$, we define its *$i$-hop shortest path neighborhood* as the set of nodes in $G$ reachable from $u$ through a shortest path of length $i$. Then, the state of $u$ is updated by separately aggregating messages from each $i$-hop neighborhood for $1 \leq i \leq k$, for some choice of $k$. This corresponds to a single iteration (i.e., layer) of SP-MPNNs, and we can use multiple layers as in MPNNs. For example, consider the graph shown in Figure 1, where 1-hop, 2-hop and 3-hop shortest path neighborhoods of the white node are represented by different colors. SP-MPNNs first separately aggregate representations from each neighborhood, and then combine all hop-level aggregates with the white node embedding to yield the new node state.

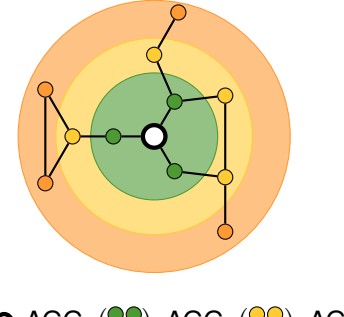

$$\mathsf{COM}\Big(\mathbf{O}, \mathsf{AGG}_1\big(\text{●●}\big), \mathsf{AGG}_2\big(\text{○○}\big), \mathsf{AGG}_3\big(\text{○○}\big)\Big)$$

**Figure 1:** SP-MPNNs update the state of the white node, by aggregating from its different shortest path neighborhoods, which are color-coded.

Our framework builds on a line of work on GNNs using multi-hop aggregation [16, 17, 24, 25], but distinguishes itself with key choices, as discussed in detail in Section 6. Most importantly, the choice of aggregating over shortest path neighborhoods ensures *distinct neighborhoods*, and thus avoids redundancies, i.e., nodes are not repeated over different hops. SP-MPNNs enable a *direct communication* between nodes in different hops, which in turn, enables more holistic node state updates. Our contributions can be summarized as follows:

— We propose SP-MPNNs, which strictly generalize MPNNs, and enable direct message passing between nodes and their shortest path neighbors. Similarly to MPNNs, our framework can be instantiated in many different ways, and encapsulates several recent models, including the state-of-the-art Graphormer [23].

— We show that SP-MPNNs can discern any pair of graphs which can be discerned either by the 1-WL graph isomorphism test, or by the shortest path graph kernel, making SP-MPNNs strictly more expressive than MPNNs which are upper bounded by the 1-WL test [12, 26].

— We present a logical characterization of SP-MPNNs, based on the characterization given for MPNNs [27], and show that SP-MPNNs can capture a larger class of functions than MPNNs.

— In our empirical analysis, we focus on a basic, simple model, called *shortest path networks*. We show that shortest path networks alleviate over-squashing, and propose carefully designed synthetic datasets through which we validate this claim empirically.

— We conduct a comprehensive empirical evaluation using real-world graph classification and regression benchmarks, and show that shortest path networks achieve state-of-the-art performance.

All proofs for formal statements, as well as further experimental details, can be found in the appendix.

## 2 Message Passing Neural Networks

*Graph neural networks (GNNs)* [8, 9] have become very prominent in graph machine learning [11–13], as they encode desirable relational inductive biases [10]. *Message-passing neural networks (MPNNs)* [14] are an effective class of GNNs, where each node $u$ is assigned an initial state vector $\mathbf{h}_u^{(0)}$, which is iteratively updated based on the state of its neighbors $\mathcal{N}(u)$ and its own state, as:

$$\mathbf{h}_u^{(t+1)} = \mathsf{COM}\Big(\mathbf{h}_u^{(t)}, \mathsf{AGG}(\mathbf{h}_u^{(t)}, \{\!\!\{\mathbf{h}_v^{(t)} \mid v \in \mathcal{N}(u)\}\!\!\})\Big),$$

where $\{\!\!\{\cdot\}\!\!\}$ denotes a multiset, and $\mathsf{COM}$ and $\mathsf{AGG}$ are differentiable *combination*, and *aggregation* functions, respectively. An MPNN is *homogeneous* if each of its layers uses the same $\mathsf{COM}$ and $\mathsf{AGG}$ functions, and *heterogeneous*, otherwise.

The choice for the aggregate and combine functions varies across models, e.g., graph convolutional networks (GCNs) [11], graph isomorphism networks (GINs) [12], and graph attention networks (GATs) [13]. Following message passing, the final node embeddings are *pooled* to form a graph embedding vector to predict properties of entire graphs. The pooling often takes the form of simple averaging, summing or element-wise maximum.

MPNNs naturally capture the input graph structure and are computationally efficient, but they suffer from several well-known limitations. MPNNs are limited in expressive power, at most matching the power of the *1-dimensional Weisfeiler Leman graph isomorphism test (1-WL)* [12, 26]: graphs cannot be distinguished by MPNNs if 1-WL does not distinguish them, e.g., the pair of graphs in Figure 2 are indistinguishable by MPNNs. Hence, several alternatives, i.e., approaches based on *unique node identifiers* [28], *random node features* [29, 30], or *higher-order* GNN models [26, 31–33], have been proposed to improve on this bound. Two other limitations, known as *over-smoothing* [34, 35] and *over-squashing* [15], are linked to using more mes-

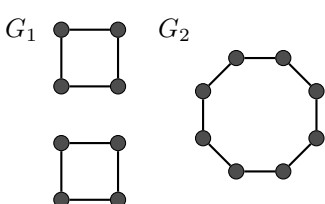

**Figure 2:** The graphs $G_1$ and $G_2$ are indistinguishable by 1-WL.

sage passing layers. Briefly, using more message passing layers leads to increasingly similar node representations, hence to over-smoothing. Concurrently, the receptive field in MPNNs grows exponentially with the number of message passing iterations, but the information from this receptive field is compressed into fixed-length node state vectors. This leads to substantial loss of information, referred to as over-squashing.

## 3  Shortest Path Message Passing Neural Networks

We consider *simple, undirected, connected*[1] graphs $G = (V, E)$ and write $\rho(u, v)$ to denote the *length of the shortest path* between nodes $u, v \in V$. The *i-hop shortest path neighborhood* of $u$ is defined as $\mathcal{N}_i(u) = \{v \in V \mid \rho(u, v) = i\}$, i.e., the set of nodes reachable from $u$ through a shortest path of length $i$. In SP-MPNNs, each node $u \in V$ is assigned an initial state vector $\mathbf{h}_u^{(0)}$, which is iteratively updated based on the node states in the *shortest path neighborhoods* $\mathcal{N}_1(u), \ldots, \mathcal{N}_k(u)$ for some choice of $k \geq 1$, and its own state as:

$$\mathbf{h}_u^{(t+1)} = \mathsf{COM}\Big(\mathbf{h}_u^{(t)}, \mathsf{AGG}_{u,1}, \ldots, \mathsf{AGG}_{u,k}\Big),$$

where $\mathsf{COM}$ and $\mathsf{AGG}_{u,i} = \mathsf{AGG}_i(\mathbf{h}_u^{(t)}, \{\!\{\mathbf{h}_v^{(t)} \mid v \in \mathcal{N}_i(u)\}\!\})$ are differentiable *combination*, and *aggregation* functions, respectively. We write SP-MPNN ($k = j$) to denote an SP-MPNN model using neighborhoods at distance up to $k = j$. Importantly, $\mathcal{N}(u) = \mathcal{N}_1(u)$ for simple graphs, and so SP-MPNN ($k = 1$) is a standard MPNN.

Similarly to MPNNs, different choices for $\mathsf{AGG}$ and $\mathsf{COM}$ lead to different SP-MPNN models. Moreover, graph pooling approaches [36], and related notions directly translate to SP-MPNNs, and so do, e.g., sub-graph sampling approaches [37, 38] for scaling to large graphs. Similarly to MPNNs, we can incorporate a *global readout* component to define SP-MPNNs with global readout:

$$\mathbf{h}_u^{(t+1)} = \mathsf{COM}\Big(\mathbf{h}_u^{(t)}, \mathsf{AGG}_{u,1}, \ldots, \mathsf{AGG}_{u,k}, \mathsf{READ}(\mathbf{h}_u^{(t)}, \{\!\{\mathbf{h}_v^{(t)} \mid v \in G\}\!\})\Big),$$

where $\mathsf{READ}$ is a permutation-invariant readout function.

To make our study concrete, we define a basic, simple, instance of SP-MPNNs, called *shortest path networks (SPNs)* as:

$$\mathbf{h}_u^{(t+1)} = \mathsf{MLP}\Big((1 + \epsilon)\,\mathbf{h}_u^{(t)} + \sum_{i=1}^{k} \alpha_i \sum_{v \in \mathcal{N}_i(u)} \mathbf{h}_v^{(t)}\Big),$$

where $\epsilon \in \mathbb{R}$, and $\alpha_i \in [0, 1]$ are learnable weights, satisfying $\alpha_1 + \ldots + \alpha_k = 1$[2]. That is, SPNs use summation to aggregate within hops, *weighted summation* for aggregation across all $k$ hops, and finally, an MLP as a combine function.

---

[1]We assume connected graphs for ease of presentation: All of our results can be extended to disconnected graphs, see the appendix for further details.

[2]When the weights are unconstrained, the model performs slightly worse and overfits. Hence, this restriction not only provides a means to interpret neighborhood importance, but also acts as an effective regularizer.

Intuitively, SPNs can directly aggregate from different neighborhoods, by weighing their contributions. It is easy to see that SPNs with $k = 1$ are identical to GIN, but observe that SPNs with arbitrary $k$ are also identical to GIN as long as the weight of the direct neighborhood is learned to be $\alpha_1 = 1$. We use SPNs throughout this paper as an *intentionally simple* baseline, as we seek to purely evaluate the impact of our extended message passing paradigm with *minimal reliance* on tangential model choices, e.g., including attention, residual connections, recurrent units, etc.

The SP-MPNN framework offers a unifying perspective for several recent models in graph representation learning using shortest path neighborhoods. In particular, SP-MPNN with global readout encapsulates models such as Graphormer[3][23], the winner of the 2021 PCQM4M-LSC competition in the KDD Cup. Indeed, Graphormer is an instance of SP-MPNNs with global readout over simple, undirected, connected graphs (without edge types), as shown in the following proposition:

**Proposition 1.** *A Graphormer with a maximum shortest path length of $M$ is an instance of SP-MPNN* ($k = M - 1$) *with global readout.*

## 4 Properties of Shortest Path Massing Passing Neural Networks

In this section, we study the properties of SP-MPNNs, and specifically analyze their information propagation properties as well as their expressive power.

### 4.1 Information Propagation: Alleviating Over-squashing

Consider a graph $G$, its *adjacency matrix* representation $\mathbf{A}$, and its *diagonal degree matrix* $\mathbf{D}$, indicating the number of edges incident to every node in $G$. We also consider variations of the degree matrix, e.g., $\tilde{\mathbf{D}} = \mathbf{D} + \mathbf{I}$, where $\mathbf{I}$ is the *identity matrix*. In our analysis, we focus on the *normalized adjacency matrix* $\hat{\mathbf{A}} = \tilde{\mathbf{D}}^{-0.5}(\mathbf{A} + \mathbf{I})\tilde{\mathbf{D}}^{-0.5}$ to align with recent work analyzing over-squashing [39].

To study over-squashing, Topping et al. [39] consider the Jacobian of node representations relative to initial node features, i.e., the ratio $\partial \mathbf{h}_u^{(r)} / \partial \mathbf{h}_v^{(0)}$, where $u, v \in V$ are separated by a distance $r \in \mathbb{N}^+$. This Jacobian is highly relevant to over-squashing, as it quantifies the effect of initial node features for distant nodes ($v$), on target node ($u$) representations, when sufficiently many message passing iterations ($r$) occur. In particular, a low Jacobian value indicates that $\mathbf{h}_v^{(0)}$ minimally affects $\mathbf{h}_u^{(r)}$.

To standardize this Jacobian, Topping et al. [39] assume the normalized adjacency matrix for $\mathsf{AGG}$, i.e., neighbor messages are weighted by their coefficients in $\hat{\mathbf{A}}$ and summed. This is a useful assumption, as $\hat{\mathbf{A}}$ is normalized, thus preventing artificially high gradients. Furthermore, a smoothness assumption is made on the gradient of $\mathsf{COM}$, as well as that of individual MPNN messages, i.e., the terms summed in aggregation. More specifically, these gradients are bounded by quantities $\alpha$ and $\beta$, respectively. Given these assumptions, it has been shown that $|\partial \mathbf{h}_u^{(r)} / \partial \mathbf{h}_v^{(0)}| \leq (\alpha\beta)^r \hat{\mathbf{A}}_{uv}^r$, upper-bounding the absolute value of the Jacobian [39]. Observe that the term $\hat{\mathbf{A}}_{uv}^r$ typically decays *exponentially* with $r$ in MPNNs, as node degrees are typically much larger than 1, imposing decay due to $\tilde{\mathbf{D}}$. Moreover, this term is *zero* before iteration $r$ due to *under-reaching*.

Analogously, we also consider normalized adjacency matrices within SP-MPNNs. That is, we use the matrix $\hat{\mathbf{A}}_i = \tilde{\mathbf{D}}_i^{-0.5}(\mathbf{A}_i + \mathbf{I})\tilde{\mathbf{D}}_i^{-0.5}$ within each $\mathsf{AGG}_i$, where $\mathbf{A}_i$ is the $i$-hop 0/1 adjacency matrix, which verifies $(\mathbf{A}_i)_{uv} = 1 \Leftrightarrow \rho(u, v) = i$, and $\tilde{\mathbf{D}}_i$ is the corresponding degree matrix. By design, SP-MPNNs span $k$ hops per iteration, and thus let information from $v$ reach $u$ in $q = \lceil r/k \rceil$ iterations. For simplicity, let $r$ be an exact multiple of $k$. In this scenario, $\partial \mathbf{h}_u^{(q)} / \partial \mathbf{h}_v^{(0)}$ is non-zero and depends on $(\hat{\mathbf{A}}_k)_{uv}^q$ (this holds by simply considering $k$-hop aggregation as a standard MPNN). Therefore, for larger $k$, $q \ll r$, which reduces the adjacency exponent substantially, thus improving gradient flow. In fact, when $r \leq k$, the Jacobian $\partial \mathbf{h}_u^{(1)} / \partial \mathbf{h}_v^{(0)}$ is only *linearly* dependent on $(\hat{\mathbf{A}}_r)_{uv}$. Finally, the hop-level neighbor separation of neighbors within SP-MPNN further improves the Jacobian, as node degrees are *partitioned* across hops. More specifically, the set of all connected nodes to $u$ is partitioned based on distance, leading to smaller degree matrices at every hop, and thus to less severe normalization, and better gradient flow, compared to, e.g, using a fully connected layer across $G$ [15].

---

[3]We follow the authors' terminology, and refer to the specific model defined using shortest path biases and degree positional embeddings as "Graphormer". This Graphormer model is introduced in detail in the appendix.

## 4.2 Expressive Power of Shortest Path Message Passing Networks

Shortest path computations within SP-MPNNs introduce a direct correspondence between the model and the shortest path (SP) kernel [40], allowing the model to distinguish any pair of graphs SP distinguishes. At the same time, SP-MPNNs contain MPNNs which can match the expressive power of 1-WL when supplemented with injective aggregate and combine functions [12]. Building on these observations, we show that SP-MPNNs can match the expressive power of both kernels:

**Theorem 1.** *Let $G_1$, $G_2$ be two non-isomorphic graphs. There exists a SP-MPNN $\mathcal{M} : \mathcal{G} \to \mathbb{R}$, such that $\mathcal{M}(G_1) \neq \mathcal{M}(G_2)$ if either 1-WL distinguishes $G_1$ and $G_2$, or SP distinguishes $G_1$ and $G_2$.*

Fundamentally, the SP and 1-WL kernels distinguish substantially different sets of graphs, and thus the theorem implies that SP-MPNNs can represent an interesting and wide variety of graphs. Indeed, SP exploits distance information between all pairs of graph nodes, and thus it (and SP-MPNN) can determine, e.g., graph connectedness, by considering whether a shortest path exists between all pairs of nodes. In contrast, 1-WL is based on iterative local hash operations, and cannot detect connectedness. To illustrate this, observe that 1-WL fails to distinguish the graphs $G_1$ and $G_2$ shown in Figure 2, whereas SP can distinguish these graphs. Since SP distinguishes a different set of graphs than 1-WL, SP-MPNNs strictly improve on the expressive power of MPNNs. For example, SP-MPNNs ($k \geq 2$) can already distinguish the graphs $G_1$ and $G_2$.

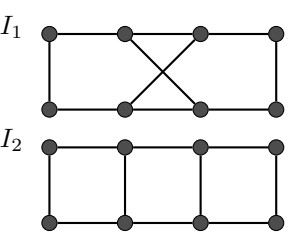

**Figure 3:** A pair of connected graphs $I_1$, $I_2$ which can be distinguished by SP, but not by 1-WL.

While checking connectedness is a somewhat obvious aspect of the power of SP, the difference between the power of the kernels SP and 1-WL goes beyond this. In fact, SP offers an expressiveness gain over 1-WL even on connected graphs. To demonstrate this, consider the simple pair of connected graphs $I_1, I_2$, shown in Figure 3. This pair of graphs is not distinguishable by 1-WL, but have different shortest path matrices. Indeed, the Wiener Indices of both graphs, i.e., the sum of the shortest path lengths, are distinct: $I_1$ has a Wiener Index of 50, whereas $I_2$ has a Wiener Index of 56. Moreover, there exist shortest paths of length 4 in $I_2$ (crossing the graph from a corner to the opposite corner), whereas no such paths exist in $I_1$. Hence, the SP kernel can distinguish $I_1$ and $I_2$.

SP exploits additional structural information, which allows it to discriminate a distinct set of graphs than 1-WL. Importantly, however, the SP kernel alone is *agnostic* to node features, and thus is unable to distinguish structurally isomorphic graphs with distinct node features. In contrast, 1-WL exploits node features, and thus can easily distinguish the aforementioned graphs. Therefore, combining the two kernels in SP-MPNN unlocks the strengths of both kernels, namely the ability of SP to capture node distances, and the feature processing of 1-WL along with its ability to capture local structures. Nonetheless, the power provided by 1-WL and SP also has limitations, as neither kernel can distinguish the graphs $H_1$ and $H_2$ shown in Figure 4. It is easy to see that SP-MPNNs cannot discern $H_1$ and $H_2$ either.

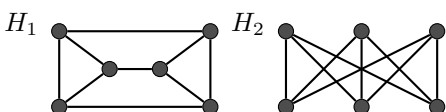

**Figure 4:** The graphs $H_1$ and $H_2$ are indistinguishable by neither 1-WL nor SP [41].

Unsurprisingly, the choice of $k$ affects expressive power. On one hand, $k = n - 1$ allows SP-MPNNs to replicate SP, whereas setting $k = 1$ reduces them to MPNNs. Also note that the expressive power of SP-MPNNs cannot be completely characterized within the WL hierarchy since, e.g., $H_1$ and $H_2$, which cannot be distinguished by SP-MPNNs, can be distinguished by folklore 2-WL. In practice, the optimal $k$ relates to the problem radius of the prediction task [15]: A higher $k$ value ($k > 1$) is not helpful for predicting a local graph property, e.g., neighbor counting, whereas tasks with long-range dependencies necessitate and benefit from a higher $k$.

## 4.3 Logical Characterization of Shortest Path Message Passing Networks

Beyond distinguishing graphs, we can study the expressive power of SP-MPNNs in terms of the *class of functions* that they can capture, following the logical characterization given by Barceló et al. [27]. This characterization is given for node classification and establishes a correspondence

between first-order formulas and MPNN classifiers. Briefly, a first-order formula $\phi(x)$ with one free variable $x$ can be viewed as a *logical node classifier*, by interpreting the free variable $x$ as a node $u$ from an input graph $G$, and verifying whether the property $\phi(u)$ holds in $G$, i.e., $G \models \phi(u)$. For instance, the formula $\phi(x) = \exists y E(x, y) \wedge Red(y)$ holds when $x$ is interpreted as a node $u$ in $G$, if and only if $u$ has a red neighbor in $G$. An MPNN $\mathcal{M}$ *captures a logical node classifier* $\phi(x)$ if $\mathcal{M}$ admits a parametrization such that for all graphs $G$ and nodes $u$, $\mathcal{M}$ maps $(G, u)$ to *true* if and only if $G \models \phi(u)$. Barceló et al. [27] show in their Theorem 5.1 that any $\mathsf{C}^2$ classifier can be captured by an MPNN with a *global readout*. $\mathsf{C}^2$ is the two-variable fragment of the logic $\mathsf{C}$, which extends first-order logic with counting quantifiers, e.g., $\exists^{\geq m} x\ \phi(x)$ for $m \in \mathbb{N}$.

It would be interesting to analogously characterize SP-MPNNs with global readout. To this end, let us extend the relational vocabulary with a distinct set of binary *shortest path predicates* $E_i$, $2 \leq i \leq k$, such that $E_i(u, v)$ evaluates to *true* in $G$ if and only if there is a shortest path of length $i$ between $u$ and $v$ in $G$. Let us further denote by $\mathsf{C}_k^2$ the extension of $\mathsf{C}^2$ with such shortest path predicates. Observe that $\mathsf{C}^2 \subsetneq \mathsf{C}_k^2$: given the graphs $G_1, G_2$ from Figure 2, the $\mathsf{C}_2^2$ formula $\phi(x) = \exists^{\geq 2} y\ E_2(x, y)$ evaluates to *false* on all $G_1$ nodes, and *true* on all $G_2$ nodes. By contrast, no $\mathsf{C}^2$ formula can produce different outputs over the nodes of $G_1, G_2$, due to a correspondence between 1-WL and $\mathsf{C}^2$ [42].

Through a simple adaptation of Theorem 5.1 of Barceló et al. [27], we obtain the following theorem:

**Theorem 2.** *Given a $k \in \mathbb{N}$, each $\mathsf{C}_k^2$ classifier can be captured by a SP-MPNN with global readout.*

### 4.4 Time and Space Complexity

In SP-MPNNs, message passing requires the shortest path neighborhoods up to the threshold of $k$ hops to be computed in advance. In the worst case, this computation reduces to computing the all-pairs unweighted shortest paths over the input graph, which can be done in $O(|V||E|)$ time using breadth-first search (BFS). Importantly, this computation is only required *once*, and the determined neighborhoods can subsequently be re-used at no additional cost. Hence, this overhead can be considered as a *pre-computation* which does not affect the online running time of the model.

Given all-pairs unweighted shortest paths, SP-MPNNs perform aggregations over a worst-case $O(|V|^2)$ elements as it considers all pairs of nodes, analogously to MPNNs over a fully connected graph. In the average case, the running time of SP-MPNNs depends on the size of nodes' $k$-hop neighborhoods, which are typically larger than their direct neighborhoods. However, this increase in average aggregation size is alleviated in practice as SP-MPNNs can aggregate across all $k$ hop neighborhoods in parallel. Therefore, SP-MPNN models typically run efficiently and can feasibly be applied to common graph classification and regression benchmarks, despite considering a richer neighborhood than standard MPNNs.

As with MPNNs , SP-MPNNs only require $O(|V|)$ node representations to be stored and updated at every iteration. The space complexity in terms of model parametrization then depends on choices for $\mathsf{AGG}_i$ and $\mathsf{COM}$. In the worst case, with $k$ distinct parametrized $\mathsf{AGG}_i$ functions, e.g., $k$ distinct neural networks, SP-MPNNs store $O(k)$ parameter sets. By contrast, using a uniform aggregation across hops yields an analogous space complexity as MPNNs.

## 5 Empirical Evaluation

In this section, we evaluate (i) SPNs and a small Graphormer model on dedicated synthetic experiments assessing their information flow contrasting with classical MPNNs; (ii) SPNs on real-world graph classification [43, 44] tasks and (iii) a basic relational variant of SPNs, called R-SPN, on regression benchmarks [45, 46]. In all experiments, SP-MPNN models achieve state-of-the-art results. Further details and additional experiments on MoleculeNet [47, 48] can also be found in the appendix. Moreover, our code can be found at http://www.github.com/radoslav11/SP-MPNN.

### 5.1 Experiment: Do all red nodes have at most two blue nodes at $\leq h$ hops distance?

In this experiment, we evaluate the ability of SP-MPNNs to handle long-range dependencies, and compare against standard MPNNs. Specifically, we consider classification based on *counting* within $h$-hop neighborhoods: *given a graph with node colors including, e.g., red and blue, do all red nodes have at most 2 blue nodes within their $h$-hop neighborhood?*

**Table 1:** Results (Accuracy) for SPNs with $k = \{1, 5\}$ on the $h$-Proximity benchmarks.

| Model | 1-Proximity | 3-Proximity | 5-Proximity | 8-Proximity | 10-Proximity |
|---|---|---|---|---|---|
| GCN | $65.0_{\pm 3.5}$ | $50.0_{\pm 0.0}$ | $50.0_{\pm 0.0}$ | $50.1_{\pm 0.0}$ | $49.9_{\pm 0.0}$ |
| GAT | $91.7_{\pm 7.7}$ | $50.4_{\pm 1.0}$ | $49.9_{\pm 0.0}$ | $50.0_{\pm 0.0}$ | $50.0_{\pm 0.0}$ |
| SPN ($k = 1$) | $\mathbf{99.4}_{\pm 0.6}$ | $50.5_{\pm 0.7}$ | $50.2_{\pm 1.0}$ | $50.0_{\pm 0.9}$ | $49.8_{\pm 0.8}$ |
| SPN ($k = 5, L = 2$) | $96.4_{\pm 0.8}$ | $94.7_{\pm 1.6}$ | $95.8_{\pm 0.9}$ | $96.2_{\pm 0.6}$ | $96.2_{\pm 0.6}$ |
| SPN ($k = 5, L = 5$) | $96.9_{\pm 0.6}$ | $\mathbf{95.5}_{\pm 1.6}$ | $\mathbf{96.8}_{\pm 0.7}$ | $96.8_{\pm 0.6}$ | $\mathbf{96.8}_{\pm 0.6}$ |
| Graphormer | $94.1_{\pm 2.3}$ | $94.7_{\pm 2.7}$ | $95.1_{\pm 1.8}$ | $\mathbf{97.3}_{\pm 1.4}$ | $96.8_{\pm 2.1}$ |

This question presents multiple challenges for MPNNs. First, MPNNs must learn to identify the two relevant colors in the input graph. Second, they must *count* color statistics in their long-range neighborhoods. The latter is especially difficult, as MPNNs must keep track of all their long-range neighbors despite the redundancies stemming from message passing. This setup hence examines whether SP-MPNNs enable better information flow than MPNNs, and alleviate over-squashing.

**Data generation.** We propose the $h$-Proximity datasets to evaluate long-range information flow in GNNs. In $h$-Proximity, we use a graph structure based on node *levels*, where (i) consecutive level nodes are pairwise fully connected, (ii) nodes within a level are pairwise disconnected. As a result, these graphs are fully specified by their level count $l$ and the level width $w$, i.e., the number of nodes per level. We show a graph pair with $l = 3, w = 3$ in Figure 5.

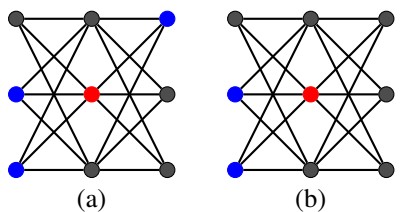

(a)        (b)

**Figure 5:** Graph (a) has one red node with *three* blue neighbors (classified as false). Graph (b) has one red node with only *two* blue neighbors (classified as true).

Using this structure, we generate pairs of graphs, classified as true and false respectively, differing only by one edge. More specifically, we generate $h$-Proximity datasets consisting each of 4500 pairs of graphs, for $h = \{1, 3, 5, 8, 10\}$. Within these datasets, we design every graph pair to be at the decision boundary for our classification task: the positive graph always has all its red nodes connected exactly to 2 blue nodes in its $h$-hop neighborhood, whereas the negative graph violates the rule by introducing one additional edge to the positive graph. We describe our data generation procedure in detail in Appendix D.

**Experimental setup.** We use two representative SP-MPNN models: SPN and a small Graphormer model. Following Errica et al. [43], we use SPN with batch normalization [49] and a ReLU non-linearity following every message passing iteration. We evaluate SPN ($k = \{1, 5\}$) and Graphormer (max distance 5) and compare with GCN [11] and GAT [13] on $h$-Proximity ($h = \{1, 3, 5, 8, 10\}$) using the risk assessment protocol by Errica et al. [43]: we fix 10 random splits per dataset, run training 3 times per split, and report the average of the best results across the 10 splits. For GCN, GAT and SPN ($k = 1$), we experiment with $T = \{1, 3, 5, 8, 10\}$ message passing layers such that $T \geq h$ (so as to eliminate any potential under-reaching), whereas we use $T = \{2, \ldots, 5\}$ for SPN ($k = 5$) and $T = \{1, \ldots, 5\}$ for Graphormer. Across all our models, we adopt the same pooling mechanism from Errica et al. [43], based on layer output addition: for $T$ message passing iterations, the pooled representation is given by $\sum_{i=1}^{T} \sum_{u \in V} \mathbf{W}_i \mathbf{h}_u^{(i-1)}$, where $\mathbf{W}_i$ are learnable layer-specific linear maps. Furthermore, we represent node colors with learnable embeddings. Finally, we use analogous hyperparameter tuning grids across all models for fairness, and set an identical embedding dimensionality of 64. Further details on hyper-parameter setup can be found in Appendix E.

**Results.** Experimental results are shown in Table 1. MPNNs all exceed 50% on 1-Proximity, but fail on higher $h$ values, whereas SPN ($k = 5$) is strong across all $h$-Prox datasets, with an average accuracy of 96.1% with two layers, and 96.6% with 5 layers. Hence, SPN successfully detects higher-hop neighbors, remains strong even when $h > k$, and improves with more layers. Graphormer also improves as $h$ increases, but is more unstable, as evidenced by its higher standard deviations. Both these findings show that SP-MPNN models relatively struggle to identify the local pattern in 1-Prox given their generality, but ultimately are very successful on higher $h$-Prox datasets. Conversely, standard MPNNs only perform well on 1-Proximity, where blue nodes are directly accessible, and

**Table 2:** Results (Accuracy) for SPN ($k = \{1, 5, 10\}$) and competing models on chemical graph classification benchmarks. Other model results reported from Errica et al. [43].

| Dataset | D&D | NCI1 | PROTEINS | ENZYMES |
|---|---|---|---|---|
| Baseline | $\mathbf{78.4}_{\pm 4.5}$ | $69.8_{\pm 2.2}$ | $\mathbf{75.8}_{\pm 3.7}$ | $65.2_{\pm 6.4}$ |
| DGCNN [54] | $76.6_{\pm 4.3}$ | $76.4_{\pm 1.7}$ | $72.9_{\pm 3.5}$ | $38.9_{\pm 5.7}$ |
| DiffPool [55] | $75.0_{\pm 3.5}$ | $76.9_{\pm 1.9}$ | $73.7_{\pm 3.5}$ | $59.5_{\pm 5.6}$ |
| ECC [56] | $72.6_{\pm 4.1}$ | $76.2_{\pm 1.4}$ | $72.3_{\pm 3.4}$ | $29.5_{\pm 8.2}$ |
| GIN [12] | $75.3_{\pm 2.9}$ | $\mathbf{80.0}_{\pm 1.4}$ | $73.3_{\pm 4.0}$ | $59.6_{\pm 4.5}$ |
| GraphSAGE [7] | $72.9_{\pm 2.0}$ | $76.0_{\pm 1.8}$ | $73.0_{\pm 4.5}$ | $58.2_{\pm 6.0}$ |
| SPN ($k = 1$) | $72.7_{\pm 2.6}$ | $\mathbf{80.0}_{\pm 1.5}$ | $71.0_{\pm 3.7}$ | $67.5_{\pm 5.5}$ |
| SPN ($k = 5$) | $77.4_{\pm 3.8}$ | $78.6_{\pm 1.7}$ | $74.2_{\pm 2.7}$ | $\mathbf{69.4}_{\pm 6.2}$ |
| SPN ($k = 10$) | $77.8_{\pm 4.0}$ | $78.2_{\pm 1.2}$ | $74.5_{\pm 3.2}$ | $67.9_{\pm 6.7}$ |

struggle beyond this. Hence, message passing does not reliably relay long-range information due to over-squashing and the high connectivity of $h$-Proximity graphs.

Interestingly, SPN ($k = 1$), or equivalently GIN, solves 1-Prox almost perfectly, whereas GAT performs slightly worse (92%), and GCN struggles (65%). This substantial variability stems from model aggregation choices: GIN uses sum aggregation and an MLP, and this offers maximal injective power. However, GAT is less injective, and effectively acts as a maximum function, which drops node cardinality information. Finally, GCN normalizes all messages based on node degrees, and thus effectively averages incoming signal and discards cardinality information.

Crucially, the basic SPN model successfully solves $h$-Prox, and is also more stable and efficient than Graphormer, since it only considers shortest path neighborhoods up to $k$, whereas Graphormer considers all-pair message passing and uses attention. Hence, SPN runs faster and is less susceptible to noise, while also being a representative SP-MPNN model, not relying on sophisticated components. For feasibility, we will solely focus on SPNs throughout the remainder of this experimental study.

## 5.2 Graph Classification

In this experiment, we evaluate SPNs on chemical graph classification benchmarks D&D [50], PROTEINS [51], NCI1 [52], and ENZYMES [53].

**Experimental setup.** We evaluate SPN ($k = \{1, 5, 10\}$) on all four chemical datasets. We also follow the risk assessment protocol [43], and use its provided data splits. When training SPN models, we follow the same hyperparameter tuning grid as GIN [43], but additionally include a learning rate of $10^{-4}$, as original learning rate choices were artificially limiting GIN on ENZYMES.

**Results.** The SPN results on the chemical datasets are shown in Table 2. Here, using $k = 5$ and $k = 10$ yields significant improvements on D&D and PROTEINS. Furthermore, SPN ($k = \{5, 10\}$) performs strongly on ENZYMES, surpassing all reported results, and is competitive on NCI1. These results are very encouraging, and reflect the robustness of the model. Indeed, NCI1 and ENZYMES have limited reliance on higher-hop information, whereas D&D and PROTEINS rely heavily on this information, as evidenced by earlier WL and SP results [57, 58]. This aligns well with our findings, and shows that SPNs effectively use shortest paths and perform strongly where the SP kernel is strong. Conversely, on NCI1 and ENZYMES, where 1-WL is strong, these models also maintain strong performance. Hence, SPNs robustly combine the strengths of both SP and 1-WL, even when higher hop information is noisy, e.g., for larger values of $k$.

## 5.3 Graph Regression

**Model setup.** We define a multi-relational version of SPNs, namely R-SPN as follows:

$$\mathbf{h}_u^{(t+1)} = (1 + \epsilon)\text{MLP}_s(\mathbf{h}_u^{(t)}) + \alpha_1 \sum_{j=1}^{R} \sum_{r_j(u,v)} \text{MLP}_j(\mathbf{h}_v^{(t)}) + \sum_{i=2}^{k} \alpha_i \sum_{v \in \mathcal{N}_i(x)} \text{MLP}_h(\mathbf{h}_v^{(t)}),$$

where $R$ is a set of relations $r_1, ..., r_R$, with corresponding relational edges $r_i(x, y)$. Essentially, R-SPN introduces multi-layer perceptrons $\text{MLP}_1, ..., \text{MLP}_R$ to transform the input with respect to

**Table 3:** Results (MAE) for R-SPN ($k = \{1, 5, 10\}$, $T = 8$) and competing models on QM9. Other model results, along with their fully adjacent (FA) extensions are as previously reported [15]. Average relative improvement by R-SPN versus the *best* GNN and FA result are shown in the last two rows.

| Property | R-GIN | | R-GAT | | GGNN | | R-SPN | | |
|---|---|---|---|---|---|---|---|---|---|
| | base | +FA | base | +FA | base | +FA | $k = 1$ | $k = 5$ | $k = 10$ |
| mu | $2.64_{\pm 0.11}$ | $2.54_{\pm 0.09}$ | $2.68_{\pm 0.11}$ | $2.73_{\pm 0.07}$ | $3.85_{\pm 0.16}$ | $3.53_{\pm 0.13}$ | $3.59_{\pm 0.01}$ | $\mathbf{2.25}_{\pm 0.17}$ | $2.32_{\pm 0.20}$ |
| alpha | $4.67_{\pm 0.52}$ | $2.28_{\pm 0.04}$ | $4.65_{\pm 0.44}$ | $2.32_{\pm 0.16}$ | $5.22_{\pm 0.86}$ | $2.72_{\pm 0.12}$ | $6.74_{\pm 0.15}$ | $1.86_{\pm 0.06}$ | $\mathbf{1.82}_{\pm 0.02}$ |
| HOMO | $1.42_{\pm 0.01}$ | $\mathbf{1.26}_{\pm 0.02}$ | $1.48_{\pm 0.03}$ | $1.43_{\pm 0.02}$ | $1.67_{\pm 0.07}$ | $1.45_{\pm 0.04}$ | $2.00_{\pm 0.01}$ | $1.27_{\pm 0.03}$ | $1.32_{\pm 0.07}$ |
| LUMO | $1.50_{\pm 0.09}$ | $1.34_{\pm 0.04}$ | $1.53_{\pm 0.07}$ | $1.41_{\pm 0.03}$ | $1.74_{\pm 0.06}$ | $1.63_{\pm 0.06}$ | $2.11_{\pm 0.02}$ | $\mathbf{1.23}_{\pm 0.03}$ | $1.26_{\pm 0.06}$ |
| gap | $2.27_{\pm 0.09}$ | $1.96_{\pm 0.04}$ | $2.31_{\pm 0.06}$ | $2.08_{\pm 0.05}$ | $2.60_{\pm 0.06}$ | $2.30_{\pm 0.05}$ | $2.95_{\pm 0.02}$ | $\mathbf{1.89}_{\pm 0.06}$ | $1.94_{\pm 0.08}$ |
| R2 | $15.63_{\pm 1.40}$ | $12.61_{\pm 0.37}$ | $52.39_{\pm 42.5}$ | $15.76_{\pm 1.17}$ | $35.94_{\pm 35.7}$ | $14.33_{\pm 0.47}$ | $22.41_{\pm 0.64}$ | $\mathbf{10.80}_{\pm 0.60}$ | $10.82_{\pm 1.30}$ |
| ZPVE | $12.93_{\pm 1.81}$ | $5.03_{\pm 0.36}$ | $14.87_{\pm 2.88}$ | $5.98_{\pm 0.43}$ | $17.84_{\pm 3.61}$ | $5.24_{\pm 0.30}$ | $29.16_{\pm 1.14}$ | $3.34_{\pm 0.16}$ | $\mathbf{2.73}_{\pm 0.05}$ |
| U0 | $5.88_{\pm 1.01}$ | $2.21_{\pm 0.12}$ | $7.61_{\pm 0.46}$ | $2.19_{\pm 0.25}$ | $8.65_{\pm 2.46}$ | $3.35_{\pm 1.68}$ | $13.39_{\pm 0.37}$ | $1.15_{\pm 0.05}$ | $\mathbf{0.96}_{\pm 0.02}$ |
| U | $18.71_{\pm 23.36}$ | $2.32_{\pm 0.18}$ | $6.86_{\pm 0.53}$ | $2.11_{\pm 0.10}$ | $9.24_{\pm 2.26}$ | $2.49_{\pm 0.34}$ | $13.61_{\pm 0.73}$ | $1.32_{\pm 0.04}$ | $\mathbf{0.96}_{\pm 0.04}$ |
| H | $5.62_{\pm 0.81}$ | $2.26_{\pm 0.19}$ | $7.64_{\pm 0.92}$ | $2.27_{\pm 0.29}$ | $9.35_{\pm 0.96}$ | $2.31_{\pm 0.15}$ | $13.65_{\pm 0.63}$ | $1.20_{\pm 0.05}$ | $\mathbf{1.02}_{\pm 0.06}$ |
| G | $5.38_{\pm 0.75}$ | $2.04_{\pm 0.24}$ | $6.54_{\pm 0.36}$ | $2.07_{\pm 0.07}$ | $7.14_{\pm 1.15}$ | $2.17_{\pm 0.29}$ | $12.22_{\pm 0.71}$ | $1.06_{\pm 0.07}$ | $\mathbf{0.94}_{\pm 0.03}$ |
| Cv | $3.53_{\pm 0.37}$ | $1.86_{\pm 0.03}$ | $4.11_{\pm 0.27}$ | $2.03_{\pm 0.14}$ | $8.86_{\pm 9.07}$ | $2.25_{\pm 0.20}$ | $5.45_{\pm 0.24}$ | $1.42_{\pm 0.05}$ | $\mathbf{1.31}_{\pm 0.03}$ |
| Omega | $1.05_{\pm 0.11}$ | $0.80_{\pm 0.04}$ | $1.48_{\pm 0.87}$ | $0.73_{\pm 0.04}$ | $1.57_{\pm 0.53}$ | $0.87_{\pm 0.09}$ | $2.90_{\pm 0.06}$ | $\mathbf{0.55}_{\pm 0.01}$ | $0.55_{\pm 0.02}$ |
| vs best GNNs: | | | | | | | $+86.3\%$ | $-50.2\%$ | $-\mathbf{51.1\%}$ |
| vs best FA models: | | | | | | | $+270\%$ | $-24.4\%$ | $-\mathbf{28.1\%}$ |

each relation, as well as a self-loop relation $r_s$, encoded by $\text{MLP}_s$, to process the updating node. For higher hop neighbors, R-SPN introduces a relation type $r_h$, encoded by $\text{MLP}_h$. R-SPN emulates the R-GIN model [45] at the first hop level, and treats higher hops as an additional edge type.

**Experimental setup.** We evaluate R-SPN ($k = \{1, 5, 10\}$) on the 13 properties of the QM9 dataset [46] following the splits and protocol (5 reruns per split) of GNN-FiLM [45]. We train using mean squared error (MSE) and report mean absolute error (MAE) on the test set. We compare R-SPN against GNN-FiLM models, as well as their fully adjacent (FA) layer variants [15]. For fairness, we only report results with $T = 8$ layers, a learning rate of 0.001, a batch size of 128 and 128-dimensional embeddings. However, we conduct a depth analysis including results with $T = \{4, 6\}$ to study the robustness of R-SPN in the appendix. Finally, due to the reported and observed instability of the original R-GIN setup (layer norm, residual connections)[45], we use the simpler pooling and update setup from SPNs with our R-SPNs.

**Results.** The results of R-SPN on all 13 properties of QM9 are shown in Table 3. In these results, R-SPN ($k = 1$) performs worse than the reported R-GIN, and this is expected given its relative simplicity, e.g., no residual connections, no layer norm. However, R-SPNs with $k = \{5, 10\}$ perform very strongly, comfortably surpassing the best MPNNs and their FA counterparts. In fact, R-SPN ($k = 10$) reduces the average MAE across all properties by over 28%. Interestingly, improvement varies across QM9 properties. On the first five properties, R-SPN ($k = 10$) yields an average relative error reduction of 8.5%, whereas this reduction exceeds 50% for U0, U, H, and G. This indicates that properties variably rely on higher-hop information, with the latter properties benefiting far more from higher $k$. All in all, these results highlight that R-SPNs not only effectively alleviate over-squashing, but also provide a strong inductive bias to improve model performance.

**Analyzing the model.** To better understand model behavior, we inspect the average learned hop weights (across 5 training runs) within the first and last layers of R-SPN ($k = 10$), $T = 8$ on the U0 property. We show the *diameter* distribution of QM9 graphs in Figure 6(a), and the learned weights in Figure 6(b).

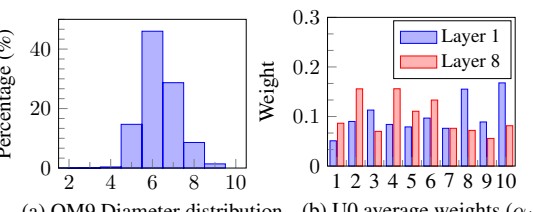

(a) QM9 Diameter distribution  (b) U0 average weights ($\alpha_i$)

**Figure 6:** Histograms for R-SPN model analysis.

Despite their small size (∼18 nodes on average), most QM9 graphs have a diameter of 6 or larger, which confirms the need for long-range information flow. This is further evidenced by the weights $\alpha_1, \ldots, \alpha_{10}$, which are non-uniform and significant for higher hops, es-

pecially within the first layer. Hence, R-SPN learns non-trivial hop aggregations. Interestingly, the weights at layers 1 and 8 are very different, which indicates that R-SPN learns sophisticated node representations, based on distinct layer-wise weighted hop aggregations. Therefore, the learned weights on U0 highlight non-trivial processing of hop neighborhoods within QM9, diverging significantly from FA layers and better exploiting higher hop information.

## 6 Related Work

The over-squashing phenomenon was first identified by Alon and Yahav [15]: applying message passing on direct node neighborhoods potentially leads to an exponentially growing amount of information being "squashed" into constant-sized embedding vectors, as the number of iterations increases. One approach to alleviate over-squashing is to "rewire" graphs, so as to connect relevant nodes (in a new graph) and shorten propagation distances to minimize bottlenecks. For instance, adding a fully adjacent final layer [45] naïvely connecting all node pairs yields substantial error reductions on QM9 [15]. DIGL [59] performs rewiring based on random walks, so as to establish connections between nodes which have small *diffusion distance* [60]. More recently, the Stochastic Discrete Ricci Flow [39] algorithm considers Ricci curvature over the input graph, where negative curvature indicates an information bottleneck, and introduces edges at negatively curved locations.

Instead of rewiring the input graphs, our study suggests better information flow for models which exploit multi-hop information through a dedicated, more general, message passing framework. We therefore build on a rich line of work that exploits higher-hop information within MPNNs [16, 17, 24, 25, 61–63]. Closely related to SP-MPNNs, the models N-GCN [16] and MixHop [17] use normalized powers of the graph adjacency matrix to access nodes up to $k$ hops away. Differently, however, these hops are *not* partitioned based on shortest paths as in SP-MPNNs, but rather are computed using *powers of the adjacency matrix*. Hence, this approach does not shrink the exponential receptive field of MPNNs, and in fact amplifies the signals coming from *highly connected* and *nearer nodes*, due to potentially redundant messages. To make this concrete, consider the graph from Figure 1: using $k = 3$ with adjacency matrix powers implies that each orange node has *one third* of the weight of a green node when aggregating at the white node. Intuitively, this is because the same nodes are repeatedly seen at different hops, which is not the case with shortest-path neighborhoods.

Our work closely resembles approaches which aggregate nodes based on shortest path distances. For instance, $k$-hop GNNs [25] compute the $k$-hop shortest path sub-graph around each node, and propagate and combine messages *inward* from hop $k$ nodes to the updating node. However, this message passing still suffers from over-squashing, as, e.g., the signal from orange nodes in Figure 1 is squashed across $k$ iterations, mixing with other messages, before reaching the white node. In contrast, SP-MPNNs enable distant neighbors to communicate *directly* with the updating node, which alleviates over-squashing significantly. Graphormer [23] builds on transformer approaches over graphs [18–20] and augments their all-pairs attention mechanism with shortest path distance-based bias. Graphormer is an instance of SP-MPNNs, and effectively exploits graph structure, but its attention still imposes a quadratic overhead, limiting its feasibility in practice. Similarly to MPNNs, our framework acts as a unifying framework for models based on shortest path message passing, and allows to precisely characterize their expressiveness and propagation properties (e.g., the theorems in Section 3 immediately apply to Graphormers).

Other approaches are proposed in the literature to exploit distant nodes in the graph, such as path-based convolution models [64, 65] and random walk approaches. Among the latter, DeepWalk [62] uses sampled random walks to learn node representations that maximize walk co-occurrence probabilities across node pairs. Similarly, random walk GNNs [61] compare input graphs with learnable "hidden" graphs using random walk-based similarity [63]. Finally, NGNNs [24], use a *nested* message passing structure, where representations are first learned by message passing within a $k$-hop rooted sub-graph, and then used for standard graph-level message passing.

## 7 Summary and Outlook

We presented the SP-MPNN framework, which enables direct message passing between nodes and their distant hop neighborhoods based on shortest paths, and showed that it improves on MPNN representation power and alleviates over-squashing. We then empirically validated this framework on the synthetic Proximity datasets and on real-world graph classification and regression benchmarks.

## Acknowledgments

The authors would like to acknowledge the use of the University of Oxford Advanced Research Computing (ARC) facility in carrying out this work. (http://dx.doi.org/10.5281/zenodo.22558)

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

# A    Proof of Proposition 1

We first recall the proposition:

**Proposition 1.** *A Graphormer with a maximum shortest path length of $M$ is an instance of SP-MPNN $(k = M - 1)$ with global readout.*

We now briefly describe the Graphormer model over simple, undirected, connected graphs without edge types. Given an input graph $G$, Graphormers perform the following steps:

1. Apply a *centrality encoding* to initial node embeddings $\mathbf{h}_u^{(0)}$. Formally, for a node $u \in G$ with degree $\deg(u)$, i.e., number of direct neighbor nodes, between 0 and a pre-set maximum degree $N$, the centrality encoding computes a refined representation $\mathbf{h}'^{(0)}_u$ as:

$$\mathbf{h}'^{(0)}_u = \mathbf{h}'^{(0)}_u + \mathbf{Z}[\deg(u)],$$

   where $\mathbf{Z} \in \mathbb{R}^{N+1 \times d}$ is a look-up embedding table, $d$ denotes the embedding dimensionality, and $\mathbf{Z}[i]$ denotes the $i^{\text{th}}$ row of $\mathbf{Z}$.

2. Iteratively update node embedding using a *spatial encoding* based on shortest path distances. Formally, for a pair of nodes $u, v \in G$ (which could be identical), an attention score function is computed using a module $\mathsf{AttScore}(\mathbf{h}_u^{(t)}, \mathbf{h}_v^{(t)})$, e.g., self-attention. Then, a *bias term*, based on the shortest path length between $u$ and $v$, $\rho(u, v)$ is obtained through a scalar look-up vector $\mathbf{b} \in \mathbb{R}^{M+1}$. Then, the attention score for a given pair of nodes is given by

$$\mathsf{AttScore'}(\mathbf{h}_u^{(t)}, \mathbf{h}_v^{(t)}) = \mathsf{AttScore}(\mathbf{h}_u^{(t)}, \mathbf{h}_v^{(t)}) + \mathbf{b}[\rho(u, v)].$$

   Note that in Graphormer, nodes with a distance greater than $M$ to $u$ are clamped to the same scalar, i.e., for $\rho(u, v) \geq M$, $\mathbf{b}[\rho(u, v)] = \mathbf{b}[M]$. Node updates are then computed by normalizing all $\mathsf{AttScore'}$ for a given node $u$ using the softmax function, and computing the following update:

$$\mathbf{h}_u^{(t+1)} = \sum_{v \in G} \mathsf{Softmax}_v\big(\mathsf{AttScore'}(\mathbf{h}_u^{(t)}, \mathbf{h}_v^{(t)})\big) \mathsf{Transform}(\mathbf{h}_v^{(t)}),$$

   where $\mathsf{Transform}$ denotes a transformation function that applies to node embeddings prior to weighted averaging, namely multiplication by a linear matrix.

*Proof.* We now reconstruct the above Graphormer using a heterogeneous SP-MPNN$(k = M - 1)$ with global readout as follows.

**Centrality encoding.** We can capture the centrality encoding through a simple first SP-MPNN layer, where aggregation functions $\mathsf{AGG}_{u,2}, \ldots, \mathsf{AGG}_{u,k}$ all return 0, and where $\mathsf{AGG}_{u,1} = \mathbf{Z}[|\mathcal{N}_1(u)|]$, i.e., we perform an analogous look-up table process to compute node degrees through the $\mathsf{AGG}_1$ component. As a result, $\mathbf{h}_u^{(1)}$ in our SP-MPNN is equivalent to $\mathbf{h}'^{(0)}_u$ in Graphormer.

**Spatial encoding.** To reconstruct the spatial encoding layer, we use an SP-MPNN layer with global readout with the following functions:

1. **Readout:**

$$\mathsf{READ}(\mathbf{h}_u^{(t)}, \{\!\!\{ \mathbf{h}_v^{(t)} \mid v \in G \}\!\!\}) = r_0 \,\|\, r_1,$$

   where $\|$ denotes the concatenation operation, $r_0 = \sum_{v \in G} e^{\mathsf{AttScore'}(\mathbf{h}_u^{(t)}, \mathbf{h}_v^{(t)} + \mathbf{b}[M])}$ is simply the scalar (i.e., $\mathbb{R}^1$) normalization constant for the softmax function and $r_1 = \sum_{v \in G} \big( e^{\mathsf{AttScore'}(\mathbf{h}_u^{(t)}, \mathbf{h}_v^{(t)} \mathbf{b}_M)} \mathsf{Transform}(\mathbf{h}_v^{(t)}) \big)$ is the un-normalized uniform attention aggregation under consistent $M$-hop bias ($r_1 \in \mathbb{R}^d$).

2. **Aggregation functions:** For $i \in \{1, \ldots, M - 1\}$

$$\mathsf{AGG}_{u,i} = a_{i,0} \,\|\, a_{i,1},$$

   where $a_{i,0} = \sum_{v \in \mathcal{N}_i(u)} e^{\mathbf{b}[i]} - e^{\mathbf{b}[M]}$ and $a_{i,1} = \sum_{v \in \mathcal{N}_i(u)} \big( (e^{\mathbf{b}[i]} - e^{\mathbf{b}[M]}) \mathsf{Transform}(\mathbf{h}_v^{(t)}) \big)$. These terms will be used by the combine function to adapt the uniform attention computed by readout to consider distance-specific biases.

3. **Combine functions:** First, the combine function computes analogous terms as the read-out and aggregation functions on $\mathbf{h}_u^{(t)}$. That is, it computes $c_0 = e^{\mathbf{b}[0]} - e^{\mathbf{b}[M]}$ and $c_1 = (e^{\mathbf{b}[0]} - e^{\mathbf{b}[M]})\mathsf{Transform}(\mathbf{h}_u^{(t)})$. Finally, the overall update is computed as follows:

$$\mathbf{h}_u^{(t+1)} = \frac{r_1 + a_{1,1} + \ldots + a_{M-1,1} + c_1}{r_0 + a_{1,0} + \ldots + a_{M-1,0} + c_0}.$$

Hence, a Graphormer model using shortest path distances up to $M$ over simple, undirected, connected graphs can be emulated by an SP-MPNN($k = M - 1$) with global readout, as required.

$\square$

## B Proof of Theorem 1

We first recall the theorem statement:

**Theorem 1.** *Let $G_1$, $G_2$ be two non-isomorphic graphs. There exists a SP-MPNN $\mathcal{M} : \mathcal{G} \to \mathbb{R}$, such that $\mathcal{M}(G_1) \neq \mathcal{M}(G_2)$ if either 1-WL distinguishes $G_1$ and $G_2$, or SP distinguishes $G_1$ and $G_2$.*

*Proof.* Let $n \in \mathbb{N}^+$ be the maximum number of nodes between $G_1$ and $G_2$. We define a heterogeneous SP-MPNN model $\mathcal{M}$ using $L = n + 1$ layers with distance parameter set to $k = n - 1$. The first layer of $\mathcal{M}$ is defined as:

$$\mathbf{h}_u^{(1)} = \mathsf{COM}^{(0)}(\mathbf{h}_u^{(0)}, \mathsf{AGG}_{u,1}^{(0)}, \ldots, \mathsf{AGG}_{u,n-1}^{(0)})$$

where $\mathbf{h}_u^{(0)}, \mathbf{h}_u^{(1)} \in \mathbb{R}^d$, $\mathsf{COM}^{(0)} : \mathbb{R}^{d+n-1} \to \mathbb{R}^d$ is an injective combination function (e.g., an MLP), and $\mathsf{AGG}_{u,i}^{(0)} = |\mathcal{N}_i(u)|$ are the aggregation functions.

All the remaining $n$ layers of $\mathcal{M}$ are defined as:

$$\mathbf{h}_u^{(t+1)} = \mathsf{COM}^{(t)}(\mathbf{h}_u^{(t)}, \mathsf{AGG}_{u,1}^{(t)}, \ldots, \mathsf{AGG}_{u,n-1}^{(t)}),$$

where $1 \leq t < n$, $\mathsf{COM}^{(t)} : \mathbb{R}^{d+n-1} \to \mathbb{R}^d$ and $\mathsf{AGG}_{u,1}^{(t)}$ are injective functions, and for each $i > 1$, $\mathsf{AGG}_{u,i}^{(t)} = 0$, i.e., the higher-hop aggregates are ignored in these layers. It is easy to see that these layers are equivalent to MPNN layers with injective functions defined as:

$$\mathbf{h}_u^{(t+1)} = \mathsf{COM}^{(t)}(\mathbf{h}_u^{(t)}, \mathsf{AGG}_{u,1}^{(t)}).$$

Intuitively, this construction encodes (1) the power of the SP kernel in the first layer of the network, and (2) the power of 1-WL using all the remaining layers in the network, which are equivalent to MPNN layers. We make a case analysis:

1. **SP distinguishes $G_1$ and $G_2$.** The SP kernel computes all pairwise shortest paths between all connected pairs of nodes in the graph and compares node-level shortest path statistics, i.e., the histograms of shortest path lengths across $G_1, G_2$ node pairs to check for isomorphism. If SP distinguishes $G_1$ and $G_2$ then there exists at least one pair of nodes with distinct shortest path histograms. Observe that the first layer of $\mathcal{M}$ yields at least one pair of distinct node representations across non-isomorphic graphs $G_1$ and $G_2$ in this case, since the diameter of each graph is at most $n - 1$ (which matches the choice of $k$), and $\mathsf{COM}$ is an injective function, acting directly on the shortest path histogram. All the remaining layers can only further refine these graphs (as these layers also define injective mappings). Finally, using an injective pooling function after $L$ iterations, we obtain $\mathcal{M}(G_1) \neq \mathcal{M}(G_2)$.

2. **1-WL distinguishes $G_1$ and $G_2$.** Observe that $\mathcal{M}$ is identical to an MPNN, excluding the very first layer, which can yield further refined node features. Hence, it suffices to show that this model is as expressive as 1-WL. This can be done by using an analogous construction to GIN (based on injective $\mathsf{AGG}$ and $\mathsf{COM}$) [12] for layers 2 to $n + 1$. In doing so, we effectively apply a standard 1-WL expressive MPNN on the more refined features provided by the first SP-MPNN layer. Notice that such a construction requires at most $n$ layers (and thus the overall SP-MPNN model would have at most $n + 1$ layers), as $n$ 1-WL iterations are sufficient to refine the node representations over graphs with at most $n$ nodes. Hence, by using a 1-WL expressive construction for SP-MPNN layers 2 to $n + 1$, and following this with an injective pooling function, we ensure that $\mathcal{M}(G_1) \neq \mathcal{M}(G_2)$ provided that 1-WL distinguishes $G_1$ and $G_2$.

Our SP-MPNN construction captures the SP kernel within its first layer by computing shortest path histograms, and ensures that node representations across $G_1$ and $G_2$ following this layer are more refined and distinct if SP distinguishes both graphs. Then, layers 2 to $n+1$ explicitly emulate a 1-WL MPNN, using injective AGG and COM functions, and apply to the more refined representations from the first layer. Therefore, these layers can distinguish the pair of graphs $G_1$ and $G_2$ if 1-WL distinguishes them. Finally, we apply an injective pooling function to maintain distinguishability. Hence, our SP-MPNN construction can distinguish $G_1$ and $G_2$ if either SP or 1-WL distinguishes both graphs, as required.

**Remark.** Note that this result easily extends to disconnected graphs. Indeed, in this scenario, one can introduce an additional aggregation over disconnected nodes. More specifically, we define an additional aggregation operation $\mathsf{AGG}_\infty$ that applies over the multiset stemming from the disconnected neighborhood $\mathcal{N}_\infty(u)$, consisting of all nodes $v \in G$ *not reachable* from $u$. Using $\mathcal{N}_\infty(u)$, the resulting SP-MPNN update in the first layer can then be written as:

$$\mathbf{h}_u^{(1)} = \mathsf{COM}^{(0)}(\mathbf{h}_u^{(0)}, \mathsf{AGG}_{u,1}^{(0)}, \ldots, \mathsf{AGG}_{u,n-1}^{(0)}, \mathsf{AGG}_{u,\infty}^{(0)}).$$

Observe that this construction is sufficient to emulate the SP kernel over disconnected graphs, as it also captures the complete histogram in this setting, including disconnected nodes. Hence, this layer is sufficient to capture the power of SP as in the original proof. Following this, the remainder of the proof is the same: $\mathsf{AGG}_{u,\infty}$ is also set to 0 within layers 2 to $n+1$.

$\square$

## C   Proof of Theorem 2

We recall the theorem statement:

**Theorem 2.** *Given a $k \in \mathbb{N}$, each $\mathsf{C}_k^2$ classifier can be captured by a SP-MPNN with global readout.*

To prove this result, we first extend the model from Barcelo et al. yielding the logical characterization to account for the additional shortest path predicates in $\mathsf{C}_k^2$.

To begin with, we first present the MPNN with global readout, known as ACR-GNN, used in the original theorem [27]. ACR-GNN is a homogeneous model, i.e., all layers are identically and uniformly parametrized. In ACR-GNN, node updates within the homogeneous layer are computed as follows:

$$\mathbf{h}_u^{(t+1)} = f\big(\mathbf{h}_u^{(t)}\mathbf{C} + (\sum_{v \in \mathcal{N}_1(u)} \mathbf{h}_v^{(t)})\mathbf{A} + (\sum_{\mathbf{v} \in V} \mathbf{h}_v^{(t)})\mathbf{R} + \mathbf{b}\big), \tag{1}$$

where $f$ is the truncated ReLU non-linearity $f(x) = \max(0, \min(x, 1))$, $\mathbf{C}, \mathbf{A}, \mathbf{R} \in \mathbb{R}^{l \times l}$ are linear maps, $\mathbf{h}_u^{(t)} \in \mathbb{R}^l$ denotes node representations and $\mathbf{b} \in \mathbb{R}^l$ is a bias vector. In this equation, $\mathbf{C}$ transforms the current node representation, $\mathbf{A}$ acts on the representations of noeds in the direct neighborhood, and $\mathbf{R}$ transforms the global readout, computed as a sum of all current node representations.

At a high level, the logical characterization of MPNNs with global readout to $\mathsf{C}^2$ is a *constructive* proof, which sets values for $\mathbf{C}, \mathbf{A}, \mathbf{R}$ and $\mathbf{b}$ so as to exactly learn the target $\mathsf{C}^2$ Boolean node classifier $\phi(x)$. This construction is *adaptive*, as the size of the MPNN depends exactly on the complexity of the formula $\phi(x)$. More specifically, the embedding dimensionality $l$ of the ACR-GNN exactly corresponds to the number of *sub-formulas* in $\phi(x)$, and the depth of the model depends on the *quantifier depth* $q$ of $\phi(x)$, which is the maximum nesting level of existential counting quantifiers. For example, the formula $\phi(x) := \exists^{\geq 2} y\big(E(x,y) \wedge \exists^{\geq 3} z\big(E(y,z)\big)\big)$ has a quantifier depth of 2.

Given a classifier $\phi(x)$, sub-formulas are traversed recursively, based on the different logical operands ($\wedge, \vee, \exists$, etc), and each assigned a dedicated embedding dimension. In parallel, entries of the learnable matrices $\mathbf{C}, \mathbf{A}, \mathbf{R}$, as well as the bias vector $\mathbf{b}$, are assigned values based on the operands used to traverse sub-formulas, so as to align with the semantics of the corresponding operands. To illustrate, consider the formula $\phi(x) = \mathrm{Red}(x) \wedge \mathrm{Blue}(x)$. This formula has 3 sub-formulas, namely (i) the Red atom, (ii) the Blue atom, and (iii) their conjunction respectively. We therefore use 3-dimensional

embeddings, and denote the corresponding dimension values for each sub-formula as $\mathbf{h}_u[1]$, $\mathbf{h}_u[2]$, and $\mathbf{h}_u[3]$ respectively. To represent the conjunction between Red and Blue (sub-formulas 1 and 2), the construction sets $\mathbf{C}_{13} = \mathbf{C}_{23} = 1$ and $\mathbf{b}_3 = -1$. This way, an ACR-GNN update only yields 1 at $\mathbf{h}_u[3]$ if $\mathbf{h}_u[1]$ and $\mathbf{h}_u[2]$ are both set to 1, in line with conjunction semantics.

Theorem 5.1 for ACR-GNNs is based on an analogous construction, but using *modal logic* operations, more specifically *modal parameters*, which are shown to be equivalent in expressive power to the logic $\mathsf{C}^2$. Modal parameters are based on the following grammar:

$$S := \mathrm{id}|e|S \cup S|S \cap S|\neg S.$$

For completeness, we now provide the same definitions as the original proof [27]. Given an undirected colored graph $G(V, E)$, the interpretation of $S$ on a node $v \in G$ is a set $\epsilon_S(v)$, defined inductively:

- if $S = \mathrm{id}$, then $\epsilon_S(v) = \{v\}$
- if $S = e$, then $\epsilon_S(v) = \{u|(u, v) \in E\}$
- if $S = S_1 \cup S_2$, then $\epsilon_S(v) = \epsilon_{S_1}(v) \cup \epsilon_{S_2}(v)$
- if $S = S_1 \cap S_2$, then $\epsilon_S(v) = \epsilon_{S_1}(v) \cap \epsilon_{S_2}(v)$
- if $S = \neg S'$, then $\epsilon_S(v) = V\ \epsilon_{S'}(v)$

The proof then uses a lemma showing that every modal logic formula can be equivalently written using only 8 different model parameters, namely: 1) id, 2) $e$, 3) $\neg e \cap \neg \mathrm{id}$, 4) $\mathrm{id} \cup e$, 5) $\neg \mathrm{id}$, 6) $\neg e$, 7) $e \cup \neg e$, 8) $e \cap \neg e$. From here, it defines precise constructions with respect to $\mathbf{A}$, $\mathbf{C}$, $\mathbf{R}$ and $\mathbf{b}$ to capture each modal parameter with respect to a counting quantifier, e.g., $\langle e \rangle^{\geq N}$.

For our purposes, we adapt this result to additionally account for the shortest path edge predicates offered by SP-MPNNs. Hence, we first propose an adapted update equation, and modify the original proof of Theorem 5.1 to incorporate the distinct edge types. For the update equation, we define learnable matrices $\mathbf{A}_i$, $i \in \{1, \ldots, k\}$ that act on neighbors within the $i$-hop neighborhood of the updating node, and accordingly instantiate the update equation of our SP-MPNN model as:

$$\mathbf{h}_u^{(t+1)} = f\Big(\mathbf{h}_u^{(t)}\mathbf{C} + \sum_i \big((\sum_{v \in \mathcal{N}_i(u)} \mathbf{h}_v^{(t)})\mathbf{A}_i\big) + (\sum_{v \in V} \mathbf{h}_v^{(t)})\mathbf{R} + \mathbf{b}\Big), \qquad (2)$$

Notice that this equation is analogous to Equation (1), with the only difference being that the single neighborhood, and the corresponding matrix $\mathbf{A}$ are replaced by $k$ neighborhoods. Using this update equation, we now lift the result of Theorem 5.1 in Barceló et al. [27] to include the additional edge predicates. To this end, we use an adapted grammar $S$, which includes $k$ edge predicates $e_1, e_2, \ldots, e_k$ (where $e_1$ is the standard edge predicate) in lieu of just $e$. Accordingly, the interpretation of these symbols is as follows:

- if $S = e_i$, then $\epsilon_S(v) := \{u|(u, v) \in E_i\}$.

By replacing $e$ with $k$ different (mutually exclusive) edge symbols $e_1, \ldots, e_k$, we obtain a modal logic defined over multiple disjoint edge types. As such, the 8 cases for the original proof must be adapted to account for the different $e_i$, leading to sub-cases with every $e_i$ for all cases including $e$ in the original proof. In particular, we now provide the construction, adapted from the original proof and corresponding to the original 8 cases, that is sufficient to represent any formula with the additional edge predicates in our setting.

In what follows, we let $\varphi_k$ denote sub-formula $k$ (which is represented using the $k^{\text{th}}$ embedding dimension, analogously to the original proof. Moreover, for ease of notation, we represent entry $kl$ in matrix $\mathbf{A}_i$ as $\mathbf{A}_{i,kl}$. The construction of our SP-MPNN model is as follows:

- *Case a.* if $\varphi_l = \langle \mathrm{id} \rangle^{\geq N}\varphi_k$, then $\mathbf{C}_{kl} = 1$ if $N = 1$ and 0 otherwise.
- *Case b.* For $i \in \{1, \ldots, k\}$, if $\varphi_l = \langle e_i \rangle^{\geq N}\varphi_k$, then $\mathbf{A}_{i,kl} = 1$ and $\mathbf{b}_l = -N + 1$.
- *Case c.* For $i \in \{1, \ldots, k\}$, if $\varphi_l = \langle \neg e_i \cup \neg id \rangle^{\geq N}\varphi_k$, then $\mathbf{R}_{kl} = 1$ and $\mathbf{C}_{kl} = \mathbf{A}_{i,kl} = -1$ and $\mathbf{b}_l = -N + 1$.
- *Case d.* For $i \in \{1, \ldots, k\}$, if $\varphi_l = \langle id \vee e_i \rangle^{\geq N}\varphi_k$, then $\mathbf{C}_{kl} = 1$ and $\mathbf{A}_{i,kl} = 1$ and $\mathbf{b}_l = -N + 1$.

- *Case e.* if $\varphi_l = \langle\neg\mathrm{id}\rangle^{\neq N}\phi_k$, then $\mathbf{R}_{kl} = 1$ and $\mathbf{C}_{kl} = -1$ and $\mathbf{b}_l = -N + 1$.

- *Case f.* For $i \in \{1, \ldots, k\}$, if $\varphi_l = \langle\neg e\rangle^{\geq N}\varphi_k$, then $\mathbf{R}_{kl} = 1$ and $\mathbf{A}_{i,kl} = -1$ and $\mathbf{b}_l = -N + 1$.

- *Case g.* For $i \in \{1, \ldots, k\}$, if $\varphi_l = \langle e \cup \neg e\rangle^{\geq N}\varphi_k$, then $\mathbf{R}_{kl} = 1$ and $\mathbf{b}_l = -N + 1$.

- *Case h.* For $i \in \{1, \ldots, k\}$, if $\varphi_l = \langle e \cup \neg e\rangle^{\geq N}\varphi_k$, then $\mathbf{R}_{kl} = 1$ and $\mathbf{b}_l = -N + 1$.

Finally, as in the original proof, all other unset values from the above cases for $\mathbf{A}_i$, $\mathbf{C}$, $\mathbf{R}$ and $\mathbf{b}$ are set to 0.

**Remark.** Note that the global readout in Equation (2) can be emulated internally within the SP-MPNN model by using an additional aggregation operation for disconnected components, i.e., distance $+\infty$, nodes, i.e., $\mathcal{N}_\infty$. More concretely, we can consider an additional aggregation operation $\mathsf{AGG}_{u,\infty}$, and then exactly capture eq. (2) using the following $\mathsf{AGG}$ definitions:

$$\mathsf{AGG}_{u,j} = \Big(\sum_{v \in \mathcal{N}_j(u)} \mathbf{h}_v\Big)(\mathbf{A}_j + \mathbf{R}) \text{ for } 1 \leq j \leq |V| - 1,$$

$$\mathsf{AGG}_{u,\infty} = \Big(\sum_{v \in \mathcal{N}_\infty(u)} \mathbf{h}_v\Big)\mathbf{R}, \text{ and}$$

$$\mathbf{h}_u^{(t+1)} = f\Big(\mathbf{h}_u^{(t)}(\mathbf{C} + \mathbf{R}) + \sum_{i=1}^{n-1} \mathsf{AGG}_{u,i} + \mathsf{AGG}_{u,\infty} + \mathbf{b}\Big).$$

# D   The $h$-Proximity Dataset

## D.1   Motivation

The evaluation of over-squashing has been studied in various earlier works [15, 39], with datasets such as Tree-NeighborsMatch [15] proposed to quantitatively measure this phenomenon.

**Limitations of Tree-NeighborsMatch.** The proposed setup in Tree-NeighborsMatch indeed evaluates information flow in the graph, but has certain undesirable properties that motivated our development of the $h$-Proximity datasets. First, Tree-NeighborsMatch uses a local classification property (number of blue neighbors) on the tree root node, and relies on information propagation only to acquire the label of a leaf node with the same number of blue nodes. Second, and most importantly, the tree structure in Tree-NeighborsMatch introduces a second implicit *exponential bottleneck* aside from information flow which could negatively bias our findings: As depth grows, *the number of leaf nodes in the tree also grows exponentially*, leading to not only the exponential decay due to over-squashing and propagating through the tree, but also an *exponential bottleneck of rival candidate classes sending information*. Hence, the model must not only receive the correct information, but also manipulate exponentially many messages from distinct nodes.

**Objectives of $h$-Proximity.** In light of these limitations, we developed the $h$-Proximity task, which has the following key desiderata:

1. A global classification property, relying on all nodes in the graph as opposed to a local property that must be transmitted to the root.

2. A *linear* dependence on the maximum hop length, as opposed to an exponential one. This allows us to build deeper graphs (e.g., 10-Prox) with linearly many nodes but exponentially growing receptive fields (stemming from the computational graph) and experiment with more realistic neighborhood configurations than trees.

Crucially, as the number of nodes is linear in the hop length, $h$-Proximity eliminates the collateral bottleneck stemming from prohibitive numbers of leaf nodes. Therefore, $h$-Proximity offers a more reliable evaluation tool for over-squashing, as any performance degradation on these datasets can more directly be attributed to the information propagation bottleneck, as opposed to the exponential amount of information being sent from exponentially many tree leaves.

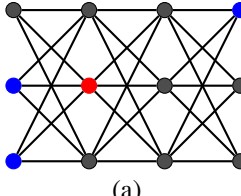 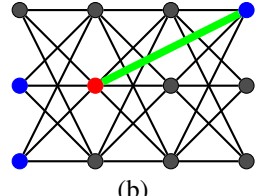

(a)             (b)

**Figure 7:** (a) A positive graph for $h = 1$ ($l = 4, w = 3$) and (b) a corresponding negative graph with an addition edge (shown in green). The red node in graph (a) has exactly two blue neighbors, but the green edge in graph (b) directly connects it to a third blue node, violating the classification objective.

### D.2 Generation Procedure

We generate all $h$-Proximity datasets in three parts. First, we generate the graph structure discussed in the main body of the paper. Then, we find a coloring of the nodes in this graph. Finally, we produce negative examples by corrupting positive graphs with an additional edge.

**Graph structure**. For every dataset, we generate 4500 graphs by sampling $l$ (the number of levels in our structure) uniformly from the discrete set $\{15, ..., 25\}$ and $w$ (the level width) from $\{3, ..., 10\}$.

**Node coloring.** We partition the 4500 graphs evenly into 3 sets of 1500 graphs, where each partition includes 1, 2, and 3 red nodes respectively, so as to produce examples with multiple red nodes, where *all* these must satisfy the classification criterion.

Given a graph and its red node allocation, we repeat the following coloring procedure until a valid coloring is found (or, alternatively, until 200 tries, at which point the graph is regenerated).

1. We select 1, 2, or 3 red nodes (depending on the partition) uniformly at random from the nodes of the input graph.

2. Given the red nodes, we identify graph nodes within the $h-$hop neighborhoods of at least one red node. We then filter out nodes which, if blue, lead to violation of the condition, i.e. a red node would have 3 or more blue neighbors in its $h$-hop neighborhood. Then, we randomly select one of the remaining nodes and color it blue. We repeat this procedure until each red node has exactly 2 blue neighbors in its $h$-hop neighborhood.

3. We randomly sample some "distant" nodes (outside the $h$-hop neighborhoods of all red nodes) to color blue. The number of selected nodes is uniformly sampled from the set $\{0, 1, 2, 3\}$. If there are insufficiently many "distant" nodes, this step is skipped.

4. We introduce 8 auxiliary colors (for a total of 10 colors) and allocate all other nodes one of these 8 colors uniformly at random.

At the end of this procedure, we obtain a graph that satisfies the classification objective, where all red nodes have exactly 2 blue nodes in their $h$-hop neighborhoods.

**Negative graph generation.** To produce negative examples from the earlier generated positive graphs, we introduce a single additional edge to make an additional "distant" blue node enter the $h$-hop neighborhood of any red node, thus violating the classification objective. Therefore, the negative graphs we produce are largely identical to the positive graphs, differing only by one additional edge. Edge addition is done as follows:

1. For every graph, identify "distant" blue nodes to one or more red nodes, and identify node pairs without an edge where an edge addition would bring a blue node within $h$ hops of a red node. Note that the node pairs need not themselves be red or blue, and could in fact be intermediary nodes offering a "shortcut".

2. Randomly sample a satisfactory edge among the aforementioned candidate edges and introduce it to the graph.

We opt for edge addition for multiple reasons. First, edge addition is fundamentally a structural modification of the graph, which affects pairwise distances in the graph. Thus, edge addition allows us to examine how the same features can propagate across the graph and offers better insights as to how these features are processed. Second, edge addition does not affect node features, and thus

**Table 4:** Diameter statistics for D&D, ENZYMES, NCI1 and PROTEINS.

| Dataset | Mean Diameter | Median Diameter |
|---|---|---|
| D&D | 19.90 | 19 |
| ENZYMES | 10.90 | 11 |
| NCI1 | 13.33 | 12 |
| PROTEINS | 11.57 | 10 |

**Table 5:** Dataset statistics for D&D, ENZYMES, NCI1, PROTEINS, and QM9.

| Dataset | #Graphs | Mean #Nodes | Mean #Edges | #Node Types | #Edge Types |
|---|---|---|---|---|---|
| D&D | 1178 | 284.3 | 815.7 | 89 | 1 |
| ENZYMES | 600 | 32.6 | 64.1 | 3 | 1 |
| NCI1 | 4110 | 29.9 | 32.3 | 37 | 1 |
| PROTEINS | 1113 | 39.1 | 72.8 | 3 | 1 |
| QM9 | 130472 | 18.0 | 18.7 | 5 | 4 |

eliminates the possibility of feature-based approximation to the task. Specifically, both positive and negative graphs have identical node features, and thus any strong model must distinguish the two from the graph structure, rather than from feature statistics.

To illustrate the negative graph generation procedure, we consider a simple example for $h = 1$, on a graph structure with $l = 4$ and $w = 3$, shown in Figure 7. In this example, we see that graph (a), the positive graph, satisfies the classification objective, as its red node is only connected to two blue nodes. Therefore, to produce a negative example, as is the case in graph (b), we add a new edge (shown in green) connecting the red node to the blue node in the rightmost level of the graph. This makes that the red node is now connected to 3 blue nodes, and thus changes the graph classification to *false*.

## E    Further Experimental Details

In this section, we provide further experimental details complementing the experimental section in the main paper.

### E.1    Hardware Configuration

We ran all our experiments on multiple identically configured server nodes, each with a V100 GPU, a 12-core Haswell CPU and 64 GB of RAM.

### E.2    Dataset Statistics

The statistics of the real-world datasets used in the experimental section of this paper, namely number of graphs, node and edge types, as well as average number of edges and nodes per graph, can be found in Table 5. We also report the mean and median graph diameter for the chemical datasets in Table 4. For the graph classification benchmarks, the number of target classes is 2 for D&D, NCI1 and PROTEINS, and 6 for ENZYMES.

### E.3    Synthetic Experiment

**Experimental protocol.** In Section 5.1, we train all models across 10 fixed splits for each $h$-Proximity dataset. On each split, we perform training three times and average the final result. Training on each split runs for 200 epochs, and test performance is computed at the epoch yielding the best validation loss.

**Hyperparameter setup.** In these experiments, we fix embedding dimensionality across all models to $d = 64$ for fairness. We also use *mean pooling* to compute graph-level outputs for all models

to keep the task challenging and enforce the learning of the long-distance target function across all nodes. Moreover, we use a node dropout with probability $0.5$ during training[4], and experiment with learning rates of $10^{-3}$ and $10^{-4}$. Furthermore, we use a batch size of 32 and adopt the same node-level pooling structure as the GIN model in the risk assessment study by Errica et al. [43] across all models. Moreover, for SPN, we additionally emulate the MLP architecture from Errica et al.: We use two-layer multi-layer perceptrons with a hidden dimension of 64 (same as the output dimensionality), such that each layer is followed by batch normalization [49] and the ReLU activation function.

**Result validation.** To validate the poor performance of MPNNs on $h$-Proximity datasets with $h \geq 3$ and discount the possibility of insufficient training, we independently trained a GAT model for 1000 epochs on one split of the 3-Proximity dataset. For this experiment, we used 3 message passing layers. We observed that it continued to struggle around 50%, similarly to what we report in the main paper. Furthermore, we trained a 300-dimensional GAT model with $T = 3$ layers on 3-Proximity for 200 epochs, and observed the same behavior. Therefore, these results confirm that the limited performance of GAT, and standard MPNNs in general, is indeed due to their structural limitations, as opposed to less accommodating hyperparameter choices.

**Discussion on MixHop.** We also sought to include MixHop as a baseline. However, this was not practically feasible, as MixHop uses normalized adjacency matrix powers, which yield dense matrices with floating-point weights for higher hops. These dense matrices make computing neighborhood aggregations computationally demanding and intractable when considering larger distances. Concretely, running an epoch of MixHop (considering hops up to 5) on all Prox datasets requires roughly 8 minutes on our hardware setup, compared to roughly 50 seconds with SPN.

In light of this issue, we exclude MixHop. Moreover, we do not compare against the default 2-hop setting of MixHop, as the resulting comparison with SPN ($k = 10$) is unfair. Nonetheless, to share some working insights, the partial experiments we could run with higher-hop MixHop showed that the model exceeds 50% training accuracy on 3, 5,8 and 10-Prox, reaching roughly about 57-58% and still improving after 200 epochs, but converged very slowly and noisily and did not exceed 51-52% test accuracy even after 200 epochs. Therefore, MixHop could potentially yield better than random performance given more training, but requires substantially more epochs and computational resources given its inherent redundancies.

### E.3.1 Additional Experiments on MoleculeNet datasets

We additionally evaluate SP-MPNN on the MoleculeNet [47] datasets. These datasets include edge features, and thus we first propose an SP-MPNN model to use this extra information.

**Model setup.** In all MoleculeNet datasets, edges are annotated with feature vectors which are typically used during message passing. Therefore, we instantiate an SP-MPNN model to use edge features analogously to the GIN implementation in the OGBG benchmarks [48]. Concretely, at the first hop level, we have tuples $(\mathbf{h}_v, \mathbf{e}_v)$ for all node neighbors, denoting the neighboring node features and the connecting edge features, respectively. Hence, we define first-hop aggregation $\mathsf{AGG}_{u,1}$ as:

$$\mathsf{AGG}_{u,1} = \sum_{v \in \mathcal{N}(u)} \mathrm{ReLU}(\mathbf{h}_v + \mathbf{e}_v).$$

Higher-hop aggregation and the overall update equation are then defined analogously to SPNs. We refer to this model as E-SPN.

**Experimental setup.** In this experiment, we use the OGB protocol on E-SPN ($k = \{1, 3, 5\}$), and compare against reported GIN and GCN results. We use 300-dimensional embeddings, follow the provided split for training, validation and testing and report average performance across 10 reruns. Furthermore, we conduct hyper-parameter tuning using largely the same grid as OGB, but additionally consider the lower learning rate of $10^{-4}$ to more comprehensively study model performance, similarly to Section 5.2. Finally, we use the full feature setup (without virtual node) from OGB and follow their feature encoding practices: We map node features to learnable embeddings at the start of message passing, and map edge features to *distinct* learnable embeddings at *every* layer.

**Results.** The results of E-SPN on MoleculeNet benchmarks are shown in Table 6. At higher values of $k$, E-SPN models yield substantial improvements on ToxCast, SIDER, ClinTox and BACE, and

---

[4]For Graphormer, we use the same default dropout mechanisms as the official repository.

**Table 6:** Results (ROC-AUC) for E-SPN and competing models on MoleculeNet graph classification benchmarks. GIN and GCN results (with features, no virtual node) are as reported in OGB [48].

| Dataset | BBBP | Tox21 | ToxCast | SIDER | ClinTox | HIV | BACE |
|---|---|---|---|---|---|---|---|
| GIN | $68.2_{\pm1.5}$ | $74.9_{\pm0.5}$ | $63.4_{\pm0.7}$ | $57.6_{\pm1.4}$ | $88.1_{\pm2.5}$ | $75.6_{\pm1.4}$ | $73.0_{\pm4.0}$ |
| GCN | $68.9_{\pm1.5}$ | $75.3_{\pm0.7}$ | $63.5_{\pm0.4}$ | $59.6_{\pm1.8}$ | $91.3_{\pm1.7}$ | $76.1_{\pm1.0}$ | $79.2_{\pm1.4}$ |
| E-SPN ($k=1$) | $\mathbf{69.1}_{\pm1.4}$ | $75.3_{\pm0.7}$ | $63.9_{\pm0.6}$ | $58.2_{\pm1.5}$ | $89.1_{\pm2.8}$ | $\mathbf{77.1}_{\pm1.2}$ | $78.3_{\pm3.0}$ |
| E-SPN ($k=3$) | $66.8_{\pm1.5}$ | $\mathbf{75.7}_{\pm1.2}$ | $64.4_{\pm0.6}$ | $59.1_{\pm1.4}$ | $\mathbf{91.8}_{\pm2.0}$ | $75.2_{\pm0.8}$ | $78.9_{\pm2.8}$ |
| E-SPN ($k=5$) | $67.5_{\pm1.9}$ | $75.4_{\pm0.8}$ | $\mathbf{65.0}_{\pm0.7}$ | $\mathbf{60.7}_{\pm0.8}$ | $88.9_{\pm1.8}$ | $76.5_{\pm1.8}$ | $\mathbf{80.9}_{\pm1.2}$ |

outperform the two baseline models. Higher-hop neighborhoods are clearly beneficial on ToxCast, BACE, and SIDER, where performance improves monotonically relative to $k$. Moreover, E-SPN models maintain strong performance on BBBP, and even yield small improvements on HIV and Tox21. These results further highlight the utility of higher-hop information, and suggest that E-SPN (as well as SPN) are promising candidates for graph classification over complex graph structures.

### E.4 Complete R-SPN Results on QM9

In this section, we present the complete results for R-SPN ($k = \{1, 5, 10\}$, $T = \{4, 6, 8\}$) on all 13 properties of the QM9 dataset. More specifically, these results are provided in Table 7, each corresponding to a QM9 property, with the best result shown in bold.

From this table, we can see that the introduction of higher-hop neighbors is key to improving the performance of R-SPN, yielding the state-of-the-art results obtained in the main paper without any additional tuning. Moreover, we notice an interesting behavior pertaining to the number of layers. Indeed, R-SPN ($k = 5$) and R-SPN ($k = 10$) are more robust with respect to the number of layers, as their performance with $T = 4$ does not drop nearly as substantially as R-SPN ($k = 1$) relative to $T = 8$. Specifically, the average error decreases by 21.6% from $T = 4$ to $T = 8$ for R-SPN ($k = 1$), but only by 7.5%, and 8.5% for $k = 5$ and $k = 10$ respectively. This suggests that using higher values of $k$ not only provides access to higher hops, but also allows this information to reach target nodes earlier on in the computation, enabling better performance with a lower number of layers.

**Table 7:** Complete results (MAE) for R-SPN with respect to the number of layers ($T$) and maximum hop size ($k$) on all properties of the QM9 dataset.

| Property | Layers | R-SPN $k = 1$ | $k = 5$ | $k = 10$ |
|---|---|---|---|---|
| mu | 4 | $4.01_{\pm 0.04}$ | $2.74_{\pm 0.15}$ | $2.68_{\pm 0.27}$ |
| | 6 | $3.66_{\pm 0.04}$ | $2.41_{\pm 0.12}$ | $2.45_{\pm 0.22}$ |
| | 8 | $3.59_{\pm 0.01}$ | $\mathbf{2.25}_{\pm 0.17}$ | $2.32_{\pm 0.20}$ |
| alpha | 4 | $9.37_{\pm 0.16}$ | $1.91_{\pm 0.04}$ | $1.84_{\pm 0.03}$ |
| | 6 | $7.07_{\pm 0.14}$ | $1.89_{\pm 0.03}$ | $\mathbf{1.82}_{\pm 0.06}$ |
| | 8 | $6.74_{\pm 0.15}$ | $1.86_{\pm 0.06}$ | $\mathbf{1.82}_{\pm 0.02}$ |
| HOMO | 4 | $2.18_{\pm 0.01}$ | $1.43_{\pm 0.02}$ | $1.46_{\pm 0.08}$ |
| | 6 | $2.05_{\pm 0.02}$ | $1.30_{\pm 0.05}$ | $1.31_{\pm 0.07}$ |
| | 8 | $2.00_{\pm 0.01}$ | $\mathbf{1.27}_{\pm 0.03}$ | $1.32_{\pm 0.07}$ |
| LUMO | 4 | $2.29_{\pm 0.02}$ | $1.33_{\pm 0.03}$ | $1.32_{\pm 0.03}$ |
| | 6 | $2.13_{\pm 0.01}$ | $1.24_{\pm 0.04}$ | $1.26_{\pm 0.04}$ |
| | 8 | $2.11_{\pm 0.02}$ | $\mathbf{1.23}_{\pm 0.03}$ | $1.26_{\pm 0.06}$ |
| gap | 4 | $3.29_{\pm 0.01}$ | $2.05_{\pm 0.05}$ | $2.06_{\pm 0.05}$ |
| | 6 | $3.02_{\pm 0.04}$ | $\mathbf{1.89}_{\pm 0.04}$ | $1.91_{\pm 0.08}$ |
| | 8 | $2.95_{\pm 0.02}$ | $\mathbf{1.89}_{\pm 0.06}$ | $1.94_{\pm 0.08}$ |
| R2 | 4 | $29.28_{\pm 0.46}$ | $12.36_{\pm 0.60}$ | $13.00_{\pm 0.60}$ |
| | 6 | $23.26_{\pm 0.59}$ | $11.44_{\pm 0.57}$ | $11.19_{\pm 0.68}$ |
| | 8 | $22.41_{\pm 0.64}$ | $\mathbf{10.80}_{\pm 0.60}$ | $10.82_{\pm 1.30}$ |
| ZPVE | 4 | $42.92_{\pm 1.62}$ | $3.25_{\pm 0.09}$ | $2.94_{\pm 0.07}$ |
| | 6 | $30.31_{\pm 1.24}$ | $3.28_{\pm 0.08}$ | $\mathbf{2.67}_{\pm 0.09}$ |
| | 8 | $29.16_{\pm 1.14}$ | $3.34_{\pm 0.16}$ | $2.73_{\pm 0.05}$ |
| U0 | 4 | $19.28_{\pm 0.77}$ | $1.21_{\pm 0.05}$ | $1.07_{\pm 0.03}$ |
| | 6 | $14.01_{\pm 0.51}$ | $1.21_{\pm 0.05}$ | $1.02_{\pm 0.05}$ |
| | 8 | $13.39_{\pm 0.37}$ | $1.15_{\pm 0.05}$ | $\mathbf{0.96}_{\pm 0.02}$ |
| U | 4 | $19.58_{\pm 0.67}$ | $1.20_{\pm 0.04}$ | $1.08_{\pm 0.05}$ |
| | 6 | $13.50_{\pm 0.51}$ | $1.18_{\pm 0.04}$ | $\mathbf{0.94}_{\pm 0.03}$ |
| | 8 | $13.61_{\pm 0.73}$ | $1.21_{\pm 0.04}$ | $0.96_{\pm 0.04}$ |
| H | 4 | $19.32_{\pm 0.42}$ | $1.24_{\pm 0.05}$ | $1.07_{\pm 0.04}$ |
| | 6 | $13.44_{\pm 0.46}$ | $1.20_{\pm 0.07}$ | $\mathbf{0.96}_{\pm 0.04}$ |
| | 8 | $13.65_{\pm 0.63}$ | $1.20_{\pm 0.05}$ | $1.02_{\pm 0.06}$ |
| G | 4 | $17.65_{\pm 0.16}$ | $1.19_{\pm 0.05}$ | $0.99_{\pm 0.03}$ |
| | 6 | $12.85_{\pm 0.43}$ | $1.12_{\pm 0.04}$ | $\mathbf{0.94}_{\pm 0.05}$ |
| | 8 | $12.22_{\pm 0.71}$ | $1.06_{\pm 0.07}$ | $\mathbf{0.94}_{\pm 0.03}$ |
| Cv | 4 | $7.53_{\pm 0.30}$ | $1.52_{\pm 0.04}$ | $1.43_{\pm 0.03}$ |
| | 6 | $5.50_{\pm 0.18}$ | $1.40_{\pm 0.02}$ | $1.41_{\pm 0.07}$ |
| | 8 | $5.45_{\pm 0.24}$ | $1.42_{\pm 0.05}$ | $\mathbf{1.31}_{\pm 0.03}$ |
| Omega | 4 | $3.29_{\pm 0.03}$ | $0.65_{\pm 0.01}$ | $0.63_{\pm 0.02}$ |
| | 6 | $3.04_{\pm 0.04}$ | $0.56_{\pm 0.01}$ | $0.56_{\pm 0.01}$ |
| | 8 | $2.90_{\pm 0.06}$ | $\mathbf{0.55}_{\pm 0.01}$ | $\mathbf{0.55}_{\pm 0.02}$ |

