# OpenReview forum: "Shortest Path Networks for Graph Property Prediction"
_logconference.io/LOG/2022/Conference — LoG 2022 Oral_

### Official Review · Reviewer_BUvT · 2022-09-25

**Overall Score:** 8
**Confidence:** 4

**Review:**

****Recommendation to accept****

########################################################################

Summary:

The paper proposes a novel message-passing framework, based on aggregating shortest-path neighbourhoods, which can break the information bottleneck and reduce over-squashing. This general framework proposed by the authors has some previous MPNNs and the Graphformer model, which has some SOTA results, as specific instances. The authors provide theoretical justification in their framework for being more expressive than MPNNs and for alleviating the over-squashing issue and show that a basic instance of the framework can perform well on various benchmarks.

########################################################################

Reasons for Score:

The method proposed has novelty and is shown to reduce the problem of over-squashing which is an important problem in the community. The theoretical and experimental justification for reducing over-squashing is well-argued and explained

########################################################################

Strengths

- The paper shows very well how SPNNs improve the problem of over-squashing, which is very important to the community in my opinion, with a theoretical justification and also with a well designed experimental justification.
- The proposed framework encompasses SOTA models including Graphformer and the connection to this model is well explained in the paper. The authors also describe how other instances of their framework can improve upon this model for some tasks. Additionally, the generality of the framework, involving shortest paths, is novel and will be beneficial to the Graph Learning Community.
- The paper is well written, the framework clearly explained, the benefits (reduced information bottleneck) well-argued and the results comprehensive. I enjoyed reading the paper.

########################################################################

Weaknesses:

- Multi-hop approaches seem the most similar to the approach outlined and the authors argue that the proposed approach improves due to avoiding redundancies (line 68) and not being computed using powers of the adjacency matrix (line 373). Due to being the most similar approach, I think it would be beneficial to either add a multi-hop approach in one of the benchmarks or to more rigorously show a problem with having these redundancies. Either way, I think the paper could be more convincing in this area.
- The value of K is an important hyper-parameter in these models, for expressive power (line 200) and on results (line 304 and line 928). I think the authors could be more clear on the effect of K (is it just dataset dependent? Is it a trend with just these three values of K in table 5 or did you see this trend continuously?). It would certainly give me a clearer idea when k=1 (Similar to standard MPNN) is limited.

########################################################################

Minor Suggestions/Thoughts

- From the second point in weaknesses, I thought it was really useful to mention the QM9 diameter distribution as it gave me a better understanding of the value of K (in terms of graph coverage). I think it would be beneficial to list the median diameter of the datasets in the benchmark tables for the value of K to make more sense to the reader.
- I thought it was strange to have the related work section so late in the paper. I think it would make more sense to describe the importance of over-squashing before your theoretical and experimental justification of your method to improve it
- I was surprised the authors created their own experiment for over-squashing rather than use the one originally proposed in [1]. Was there a reason for this? Maybe including this experiment would go further in validating your approach

[1] On The Bottleneck of Graph Neural Networks and Its Practical Applications, Uri Alon, Eran Yahav, ICLR 2021

---

### Official Review · Reviewer_1vAy · 2022-10-20

**Overall Score:** 6
**Confidence:** 3

**Review:**

This paper targets the imformation bottleneck problem of MPNN, and formulates a general framework of shortest path message passing neural networks. Some of existing GNNs can be instances of this framework. Based on this framework, the authors analyzed 1. how GNNs under this framework better handle the over-squashing problem. 2. The expressivity of shortest path message passing networks from both the ability of distinguishing graphs and the ability to capture different classes of functions. Extensive experiments are conducted from different perspectives (e.g. how well the model capture long-range dependency) to demonstrate the efficiency of the proposed framework.

Overall, although the proposed SP-MPNN is not a novel framework by its structural design, this paper provides some theoretical insights on analyzing GNNs under this framework, as well as thorough experiments to justify the statements from different perspectives with different graph learning tasks. Therefore, I think this paper has certain contribution and can be accepted.

Minor problems and questions:
1. In Table 2, the performance of SPN has no superiority on most datasets. Does this imply that the information bottleneck is not a challenge in graph prediction tasks therefore SPN has no advantages in such tasks?
2. Theorem 1 does not look like a theorem.

---

### Official Review · Reviewer_A7Nq · 2022-10-20

**Overall Score:** 8
**Confidence:** 4

**Review:**

Summary:

The authors propose a multi-hop GNN based on shortest path. An analysis of the proposed model is conducted, partly based on prior analysis of 1-hop MPNNs.
Several experiments on synthetic and real world datasets are conducted to demonstrate the efficacy of the proposed model.

Strong points:

1. The paper is well motivated and is mostly easy to follow and understand.
2. The analysis of the method sheds lights on its success in the experimental section.
3. The experimental results show improvement compared to 1 hop and transformer based methods.

Weak points:

1. Choosing the shortest path is the core of this work. However, I could not understand from the text how such a path is chosen. The best I could find in the text was in the Appendix stating that the paths can be generated in a pre-processing fashion, but it still does not tell me how the paths are generated.  I think that shortest paths being the core of this work, and given the length of the paper, it should be clearly and formally described within the main text.
I assume that the shortest path generation policy can greatly affect the obtained results. For example, do you consider the learned features to determine lengths? How do you deal with multiple shortest paths? and more.

2. One of the claims of the authors is that over-smoothing and over-squashing, which often occur in GNNs, hinder the obtained accuracy. However, besides Table 6 in the Appendix, I do not see a proper depth study of the model. I think that more experiments with deeper networks can benefit the paper.

3. In general and also with respect to my previous point, I think that the authors should conduct more typical experiments such as node classification, as is typically done in the over-smoothing and over-squashing related papers (e.g., GCNII, EGNN, PDE-GCN, GraphCON)

4. The authors are missing a discussion of a few recent papers that also consider multi-hop aggregation based GNN. For example:

"Path Integral Based Convolution and Pooling for Graph Neural Networks"

"pathGCN: Learning General Graph Spatial Operators from Paths"

Recommendation: Weak accept.

The paper discusses an important aspect of GNNs and extends the work in that direction. The authors provide an analysis of the model and conduct several experiments. In light of this and given the weak points stated above, I tend to (weakly) accept this paper.

Questions / suggestions to the authors:
1. In line 135, you state that the sum of alphas has to be 1. Can you please discuss the effect of, or provide experimental results when this constraint is not enforced?

2.In your experiments you consider hops of size 1,5 and 10. Can you please provide results with more values of k? for instance, it would be interesting to know what is the impact of k=2,3 or a significantly larger k, like 50 or 100.

3.In lines 367-368 you state that your proposition is 'better information flow for models which exploit multi-hop information through a dedicated, more general, message passing framework'. However besides the comparison with a transformer based network, I do not see a  comparison with other multi-hop methods. Can you please clarify this claim?

---

### Meta-Review · Area_Chair_khXp · 2022-11-08

**Confidence:** 4
**Recommendation:** Accept for spotlight

**Meta Review:**

The paper proposed message passing in GNNs based on the shortest-path-distance between nodes, to prevent over-squashing.
Instead of communicating only with direct neighbors, every node's state is also updated by considering k-hop nodes and the distance to these nodes.

The reviewers have highlighted the extensiveness of the experimental section, the impressive results, the thorough analysis, the useful discussion regarding the theoretical expressivity (and the comparison to WL tests), and the high quality of the writing.

I agree with all mentioned strengths, and I think that the paper provides a simple and effective approach to improving existing GNNs.
I also think that preventing over-squashing is of high interest to the community, and I recommend acceptance.

---

### Decision · Program_Chairs · 2022-11-22

Accept (Oral)